# Interpreting Unsupervised Anomaly Detection in Security via Rule Extraction

**Ruoyu Li**[§†]**, Qing Li**[∗†]**, Yu Zhang**[§]**, Dan Zhao**[†]**, Yong Jiang**[♮†]**, Yong Yang**[‡]

[§]Tsinghua University, China; [†]Peng Cheng Laboratory, China
[♮]Tsinghua Shenzhen International Graduate School, China
[‡]Tencent Security Platform Department, China
{liry19,yu-zhang23}@mails.tsinghua.edu.cn; {liq,zhaod01}@pcl.ac.cn
jiangy@sz.tsinghua.edu.cn; coolcyang@tencent.com

## Abstract

Many security applications require unsupervised anomaly detection, as malicious data are extremely rare and often only unlabeled normal data are available for training (i.e., zero-positive). However, security operators are concerned about the high stakes of trusting black-box models due to their lack of interpretability. In this paper, we propose a post-hoc method to globally explain a black-box unsupervised anomaly detection model via rule extraction. First, we propose the concept of *distribution decomposition rules* that decompose the complex distribution of normal data into multiple compositional distributions. To find such rules, we design an unsupervised Interior Clustering Tree that incorporates the model prediction into the splitting criteria. Then, we propose the Compositional Boundary Exploration (CBE) algorithm to obtain the *boundary inference rules* that estimate the decision boundary of the original model on each compositional distribution. By merging these two types of rules into a rule set, we can present the inferential process of the unsupervised black-box model in a human-understandable way, and build a surrogate rule-based model for online deployment at the same time. We conduct comprehensive experiments on the explanation of four distinct unsupervised anomaly detection models on various real-world datasets. The evaluation shows that our method outperforms existing methods in terms of diverse metrics including fidelity, correctness and robustness.

## 1 Introduction

In recent years, machine learning (ML) and deep learning (DL) have revolutionized many security applications such as network intrusion detection [1–3] and malware identification [4, 5] that outperform traditional methods in terms of accuracy and generalization. Among these works, unsupervised anomaly detection becomes more promising, which detects malicious activities by the deviation from normality. Compared to supervised methods, this type of method is more desirable in security domains as 1) it hardly requires labeled attack/malicious data during the training (i.e., zero-positive learning), which are typically much more sparse and difficult to obtain in contrast with benign data; 2) it does not fit any known threats, enabling better detection on unforeseen anomalies.

Due to the black-box nature of these models, ML/DL models are usually not directly interpretable and understandable. Many local explanation methods [6–10] have attempted to interpret the models by presenting feature importance of the decision for a single point. However, globally explaining black-box models, especially using rule extraction to characterize the whole decision boundaries, is particularly desirable in security systems since it can provide the following benefits:

---

[∗]Corresponding author: Qing Li.

37th Conference on Neural Information Processing Systems (NeurIPS 2023).

**Trust over High Stakes.** To minimize the chance of errors and potential losses, security operators tend to trust human-understandable rules rather than unintuitive outputs such as labels and numeric values from complex and incomprehensible black-box models.

**Online Defense.** Interpreting black-box models into rules with high fidelity enables easy integration with the majority of defense tools that use rule-based expressions (e.g., iptables [11], Snort [12]), thus allowing for deployment of online defense with extraordinary efficiency (e.g., Tbps throughput [13]).

Existing global ML/DL explanation approaches are mostly proposed for supervised models. Little research has been done on explaining unsupervised anomaly detection, which faces several challenges:

**Unlabeled One-class Data (CH1).** Supervised explanation methods [14–16] need labeled data of both positive and negative classes to determine the decision boundaries of black-box models. This requirement goes against unsupervised anomaly detection's advantage of not requiring attack data.

**Lack of Surrogate Models (CH2).** Global methods typically use an inherently explainable model to mimic the black-box model, such as decision trees [15, 16] and linear models [17], which are all supervised models. However, there lacks a proper surrogate model that is unsupervised and can satisfy the properties of being self-explained and well-performed for high-stake security applications.

**Accuracy Loss (CH3).** A common problem with global methods is that the surrogate model suffers from the loss of accuracy since it simplifies the original model [18]. In this case, though these methods can provide model explanation, they cannot meet the need of online deployment which requires high detection accuracy in security applications.

We observe that an important reason why simple surrogate models are ineffective is that they cannot learn well about the complex data distribution in high-dimensional space. Specifically, even one-class data may be multimodal, i.e., the overall distribution consists of multiple compositional distributions. For example, the normal activities of a host may consist of many services (e.g., web, database) and are for various purposes, and they could present considerable differences in feature space.

In light of this, this paper proposes an accurate and efficient divide-and-conquer method to globally interpret unsupervised anomaly detection. First, we propose the concept of *distribution decomposition rules* that cut the feature space into multiple subspaces, each of which encloses a compositional distribution. To obtain such rules, we design a new tree model called Interior Clustering Tree that extends the CART decision tree in terms of splitting criteria and can fully work in an unsupervised manner. Second, we propose the Compositional Boundary Exploration algorithm to obtain the *boundary inference rules* that estimate the decision boundary of the original model on each compositional distribution. To accurately and efficiently find such rules, this algorithm starts from hypercube-shaped rules and uses an approximation for gradient ascent of model prediction to find the optimal direction of iterations. By merging the distribution decomposition rules and the boundary inference rules into a rule set, we can present the inferential process of the black-box model in a human-understandable way, and build a surrogate rule-based model for online defense at the same time.

During the experiment, we use four different unsupervised anomaly detection models well trained on three benchmark datasets, and evaluate our method with five distinct explanation methods as baselines. The experiment shows that our method can extract interpretable and accurate rules from black-box models. The extracted rules outperform prior work in terms of diverse metrics including fidelity, robustness, true positive rate and true negative rate, meeting the demand of improving human trust in black-box models and maintaining high detection accuracy for online deployment. Our code is available at `https://github.com/Ruoyu-Li/UAD-Rule-Extraction`.

## 2   Related Work

To combat persistent emergence of new attacks in cyberspace, recent security applications [1–3, 19–21] make heavy use of unsupervised models to detect unknown anomalies, such as one-class classifiers [22–24], Isolation Forests [25, 26], autoencoders and variational autoencoders [27]. Despite many unsupervised model-based approaches have achieved good detection rates, security operators are still concerned about the semantic gap between black-box model prediction and human understanding, considering the risks of the great cost incurred by bad decisions [10]. To resolve such concerns, explainable AI (XAI) has been applied to anomaly detection [28–30]. For example, Kauffmann et al. propose a decomposition method to explain anomalies of one-class SVMs [28]. Philipp et al. present

an explainable deep one-class classification method called Fully Convolutional Data Description [30]. However, these methods are specific to a limited range of models and not versatile enough to accommodate the vastly heterogeneous models of unsupervised anomaly detection.

Some prior work also incorporates popular model-agnostic explanation methods, such as LIME [6], SHAP [7] and their variations [8], and applies them to explain unsupervised models [31–33] and security applications [9, 34, 35, 10]. These methods typically use sparse linear models to interpret predictions by estimating feature importance. Guo et al. propose a method named LEMNA that uses a fused lasso to explain malware classification [9]. Sipple uses Integrated Gradients [36] to attribute anomalies of IoT device failure [34]. Nonetheless, these methods can only interpret one data point at a time (i.e., local explanation) but not reveal the complete decision-making process of a model.

To fully understand how black-box models work and safely deploy them, the most appropriate method is *model-agnostic global post-hoc* explanation. It aims to match the predictions of any well-trained models with an inherently interpretable explainer, such as decision trees [15, 16], symbolic rules [14], sparse linear models [17] and decision lists [37]. In [15], the authors construct global explanations of complex black-box models in the form of a decision tree approximating the original model. Jacobs et al. propose a framework that takes an existing ML model and training dataset and generates tree models to interpret security-related decisions [16]. However, most of these methods are only suitable for interpreting supervised models that have labeled data of all classes, which are often unavailable. Though work like [38] can extract rules from unsupervised anomaly detection models, it still assumes that enough outliers exist in the training dataset judged by the black-box model so as to determine its decision boundary. This assumption may not hold in practice if a model has great generalization and can achieve a low false positive rate on normal data (e.g., [1]).

Some recent studies aggregate several local explanation models into near-global explanation [39–41]. However, this type of method is inherently computationally challenging when data volumes are large and has to make trade-offs between fidelity and coverage. While techniques like knowledge distillation can also realize model transformation to reduce complexity and promote interpretability [42, 43], the fundamental purpose of these efforts is to compress models while ensuring accuracy rather than explaining the original models with high fidelity.

## 3 Overview

### 3.1 Problem Definition

Let $\mathcal{X} \subseteq \mathbb{R}^d$ be the variable space of $d$-dimensional features; $\boldsymbol{x}$ and $x_i$ denote a data sample and the $i$-th dimension of the data sample. We give the following definitions for the rest of the paper:

**Definition 1 (Unsupervised Anomaly Detection).** *Given unlabeled negative data (i.e., normal data) $\boldsymbol{X}$ sampled from a stationary distribution $\mathcal{D}$ for training, an unsupervised model estimates the probability density function $f(\boldsymbol{x}) \approx \mathrm{P}_{\mathcal{X} \sim \mathcal{D}}(\boldsymbol{x})$, and detects an anomaly via a low probability $f(\boldsymbol{x}) < \varphi$, where $\varphi > 0$ is a threshold determined by the model itself or by humans.*

It is noted that the threshold $\varphi$ is a non-zero value, meaning that the model inevitably generates false positives, which is a common setting in most of the works [1–3] even though the false positive rate can be very low. Besides, the normal data may occasionally be contaminated or handled with errors. We consider the anomaly detection tolerant of noisy data, but their proportion in training dataset is small and we do not have any ground truth labels of the training data.

**Definition 2 (Global Explanation by Rule Extraction).** *Given a trained model $f$ with its anomaly threshold $\varphi$ and the training set $\boldsymbol{X}$, we obtain an in-distribution rule set $\mathcal{C} = \{C_1, C_2, ...\}$ that explains how the model $f$ profiles the distribution of normal data. A rule $C = ... \wedge (x_i \odot v_i) \wedge ... \wedge (x_j \odot v_j)$ is a conjunction of several axis-aligned constraints on a subset of the feature space, where $v_i$ is the bound for the $i$-th dimension and $\odot \in \{\leq, >\}$.*

Let $\boldsymbol{x} \in C$ indicate that a data sample satisfies a rule. From $\mathcal{C}$, we can build a surrogate model $h_{\mathcal{C}}(\boldsymbol{x})$, whose inference is to regard a data sample that cannot match any of the extracted rules as anomalous:

$$h_{\mathcal{C}}(\boldsymbol{x}) = \neg(\boldsymbol{x} \in C_1) \wedge \neg(\boldsymbol{x} \in C_2) \wedge ..., \ C_i \in \mathcal{C}. \tag{1}$$

**Our Goal.** We expect the extracted rules to have a high fidelity to the original model, that is, a similar coverage of normal data (i.e., true negative rate), and a similar detection rate of anomalies (i.e., true

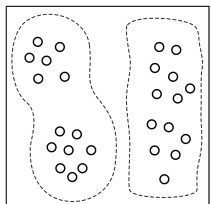 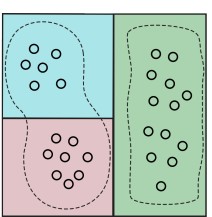 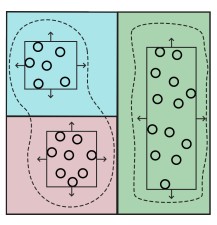 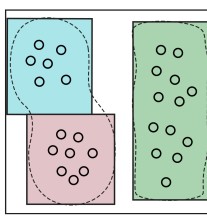

(a) The unlabeled data    (b) Compositional distribu-    (c) Process of the CBE al-    (d) The final rule set
tions                      gorithm

Figure 1: A high-level illustration of our method. The small circles are unlabeled normal data. The dashed curves are the decision boundary of the black-box model. The vertical/horizontal lines in (b) and (c) are the distribution decomposition rules.

positive rate). To this end, we formulate our objective as follow:

$$\arg\min_{\mathcal{C}} \mathcal{L}_{\mathcal{X}\sim\mathcal{D}}(\mathcal{C}, f, \varphi) + \mathcal{L}_{\mathcal{X}\not\sim\mathcal{D}}(\mathcal{C}, f, \varphi). \tag{2}$$

## 3.2 Methodology Overview

To minimize the first item in Equation (2), suppose the training data $\boldsymbol{X}$ can well represent the distribution $\mathcal{D}$, a straightforward approach is to find the bound of $\boldsymbol{X}$ as rules, such as using a hypercube to enclose the data samples which can easily achieve the minimization of the partial loss $\mathcal{L}_{\boldsymbol{x}\in\boldsymbol{X}}(\mathcal{C}, f, \varphi) = 0$. However, as $\mathcal{D}$ is not a prior distribution and we do not have labeled abnormal samples, the second item $\mathcal{L}_{\mathcal{X}\not\sim\mathcal{D}}(\mathcal{C}, f, \varphi)$ is neither deterministic nor estimable unless we create sufficient random samples and query $f$, which is challenging given the high-dimensional space of $\mathcal{X}$.

As prior studies [21, 34] suggest, normal data are typically multimodal, i.e., the overall distribution is formed by multiple compositional distributions. For example, a server supports multiple services such as web, email and database. The representations of these services can be disparate and located in different regions in feature space with little transition between the regions, making it infeasible to find a uniform rule set to accurately estimate the original model. An example is illustrated in Figure 1a.

Based on this intuition, we propose a divide-and-conquer approach. First, we propose an *Interior Clustering Tree* model (Section 4) to find the *distribution decomposition rules*, which cut the feature space into subspaces so that each subspace contains data belonging to the same compositional distribution, as shown in Figure 1b. Then, we design a *Compositional Boundary Exploration* algorithm (Section 5) to explore the decision boundary on each compositional distribution, as depicted in Figure 1c. Particularly, the algorithm starts from the minimal hypercube that encloses all data of the distribution, and finds the boundary by recursively extending the boundary following the optimal direction guided by a gradient approximation. Upon obtaining the decision boundary of a distribution, the corresponding *boundary inference rule* can be extracted. Last, the rule set that globally approximates the original model can be obtained by merging the distribution decomposition rule and the boundary inference rule of each compositional distribution, as illustrated in Figure 1d. We formally define the distribution decomposition rule and the boundary inference rule as follows.

**Definition 3 (Distribution Decomposition Rule).** *Denoted by $C_k^I$ that decomposes the overall distribution of normal data $\mathcal{D}$ into $K$ compositional distributions, i.e., $\mathrm{P}_{\mathcal{X}\sim\mathcal{D}}(\boldsymbol{x}) = \sum_{k=1}^{K} \phi_k \cdot \mathrm{P}_{\mathcal{X}\sim\mathcal{D}_k}(\boldsymbol{x}|\boldsymbol{x} \in C_k^I)$ where $\phi_k$ denotes the weight of each compositional distribution, so that a data sample $\boldsymbol{x} \sim \mathcal{D}_k$ has significantly small probability of belonging to other distributions.*

**Definition 4 (Boundary Inference Rule).** *Denoted by $C_k^E$ that estimates the decision boundary of the original model for each distribution $\mathcal{D}_k$, i.e., $\arg\min_{C_k^E} \mathcal{L}_{\mathcal{X}\sim\mathcal{D}_k}(C_k^E, f, \varphi) + \mathcal{L}_{\mathcal{X}\not\sim\mathcal{D}_k}(C_k^E, f, \varphi)$.*

With the definition of these two types of rules, we translate the objective in Equation (2) to the following objective as our intuition indicates. We give a proof of this proposition in the appendix.

**Proposition 1.** *The original objective can be estimated by finding the union of the conjunction of distribution decomposition rules and boundary inference rules for each compositional distribution:*

$$\bigcup_{k=1}^{K} \arg\min_{C_k} \mathcal{L}_{\mathcal{X} \sim \mathcal{D}_k}(C_k, f, \varphi) + \mathcal{L}_{\mathcal{X} \nsim \mathcal{D}_k}(C_k, f, \varphi), \text{where } C_k = C_k^I \wedge C_k^E. \tag{3}$$

## 4 Interior Clustering Tree

To obtain distribution decomposition rules, we first need to decide how to define a compositional distribution $\mathcal{D}_k$. We notice that, though we do not have labeled data, we can use the output of the original model that estimates the overall distribution $\mathcal{D}$ as a criterion for decomposition.

**Proposition 2.** *If two data samples $\boldsymbol{x}^{(i)}$ and $\boldsymbol{x}^{(j)}$ belong to the same distribution $\mathcal{D}_k$, the difference of their probabilities belonging to $\mathcal{D}$ will be less than $\epsilon$, where $\epsilon$ is a small constant.*

*Proof.* Recall the definition of distribution decomposition rules and compositional distributions. Since the two data samples belong to the same distribution $\mathcal{D}_k$, the probability of belonging to other distributions $\mathrm{P}_{\mathcal{X} \sim \mathcal{D}_l}(\boldsymbol{x})$ is near to zero for $l \neq k$. Hence, the probability $\mathrm{P}_{\mathcal{X} \sim \mathcal{D}}(\boldsymbol{x})$, which is the weighted sum of the probability of belonging to all the compositional distributions, is approximately equal to $\mathrm{P}_{\mathcal{X} \sim \mathcal{D}_k}(\boldsymbol{x})$ for both the data samples.

Based on this, we propose a tree-based model dubbed Interior Clustering Tree (IC-Tree), which extends the CART decision tree [44]. The main difference between IC-Tree and CART is that, rather than splitting data based on ground truth labels, IC-Tree uses the probability output by the original model as splitting criteria, enabling it to work in a completely unsupervised manner.

**Node Splitting.** Given the data $\boldsymbol{N}$ at a tree node, we first obtain the output of the anomaly detection model $f(\boldsymbol{x})$ for $\boldsymbol{x} \in \boldsymbol{N}$. Similar to decision trees, the node of an IC-Tree finds a splitting point $s = (i, b_i)$ that maximizes the gain:

$$s = \arg\max_{s} I(\boldsymbol{N}) - \frac{|\boldsymbol{N}_l|}{|\boldsymbol{N}|} I(\boldsymbol{N}_l) - \frac{|\boldsymbol{N}_r|}{|\boldsymbol{N}|} I(\boldsymbol{N}_r), \tag{4}$$

where $b_i$ is the splitting value for the $i$-th dimension, $\boldsymbol{N}_l$ and $\boldsymbol{N}_r$ are the data split to the left and right child nodes, $|\boldsymbol{N}|$ denotes the number of data samples, and $I$ is a criterion function such as Gini index $I = 2p(1-p)$ for binary classification with the probability of $p$. Specifically, we let $p$ be the average output of the anomaly detection model, which can be interpreted as the expectation of the probability that the data belong to the same distribution:

$$p = \mathbb{E}_{\boldsymbol{x} \in \boldsymbol{N}}[\mathrm{P}_{\mathcal{X} \sim \mathcal{D}}(\boldsymbol{x})] = \frac{1}{|\boldsymbol{N}|} \sum_{\boldsymbol{x} \in \boldsymbol{N}} f(\boldsymbol{x}). \tag{5}$$

An IC-Tree continues to split nodes until it satisfies one of the following conditions: i) the number of data samples at the node $|\boldsymbol{N}| = 1$; ii) for any two of the data samples at the node $\forall \boldsymbol{x}^{(i)}, \boldsymbol{x}^{(j)} \in \boldsymbol{N}$, $|f(\boldsymbol{x}^{(i)}) - f(\boldsymbol{x}^{(j)})| < \epsilon$; iii) it reaches a maximum depth $\tau$, which is a hyperparameter.

**Distribution Decomposition Rule Extraction.** A trained IC-Tree that has $K$ leaf nodes ($K \leq 2^\tau$) represents $K$ distributions separated from the overall distribution $\mathcal{D}$. Suppose the $k$-th leaf node has a depth of $\tau'$. A distribution decomposition rule that describes the $k$-th compositional distribution can be extracted by the conjunction of the splitting constraints from the root to the leaf node:

$$C_k^I = (x_i \odot_1 b_i | s_1 = (i, b_i)) \wedge ... \wedge (x_j \odot_{\tau'} b_j | s_{\tau'} = (j, b_j)), \tag{6}$$

where $\odot$ is "$\leq$" if the decision path goes left or "$>$" if the decision path goes right.

## 5 Compositional Boundary Exploration

To accurately find the decision boundary of a detection model within each compositional distribution, we propose the Compositional Boundary Exploration (CBE) algorithm (described in Algorithm 1). The CBE algorithm uses the minimal hypercube that encloses the normal data of each compositional distribution as a starting point. Further, we refer to adversarial attacks [45] and propose a method to

approximate the optimal direction to explore the decision boundary, which makes the algorithm more efficient and accurate to estimate the decision boundary.

**Starting from Hypercube (line 1).** Let $\boldsymbol{X}_k$ denote the training data falling into the $k$-th leaf node of an IC-Tree that represents a compositional distribution. Recall the definition of boundary inference rules that target $\min \mathcal{L}_{\mathcal{X} \sim \mathcal{D}_k}(C_k^{\bar{E}}, f, \varphi) + \mathcal{L}_{\mathcal{X} \nsim \mathcal{D}_k}(C_k^{E}, f, \varphi)$. We use the minimal hypercube $H_k$ as a starting point of boundary inference rules to bound every dimension of the data samples in $\boldsymbol{X}_k$ judged by the original model as normal, which obviously achieves $\mathcal{L}_{\boldsymbol{x} \in \boldsymbol{X}_k}(H_k, f, \varphi) = 0$. The minimal hypercube is enclosed by $2 \times d$ axis-aligned hyperplanes, which can be characterized by the following rule:

$$H_k = (v_1^- \leq x_1 \leq v_1^+) \wedge ... \wedge (v_d^- \leq x_d \leq v_d^+), \tag{7}$$

where $v_i^- = \min(x_i | f(\boldsymbol{x}) > \varphi, \boldsymbol{x} \in \boldsymbol{X}_k)$ and $v_i^+ = \max(x_i | f(\boldsymbol{x}) > \varphi, \boldsymbol{x} \in \boldsymbol{X}_k)$.

**Explorer Sampling (line 4~6).** The CBE algorithm explores the decision boundary of the original model by estimating the bound of one feature dimension at a time. For $i$-th dimension, we uniformly sample $N_e$ data points on each hyperplane of the hypercube, i.e., $\boldsymbol{e}^{(1)}, ..., \boldsymbol{e}^{(N_e)} \in H_k \wedge (x_i = v_i), v_i \in \{v_i^-, v_i^+\}$, which are called the *initial explorers* for this hyperplane. For an initial explorer $\boldsymbol{e}$, we further sample $N_s$ *auxiliary explorers* near it from a truncated multivariant Gaussian distribution denoted by $\mathcal{N}(\boldsymbol{e}, \boldsymbol{\Sigma}, i)$. Particularly, the center of sampling is the explorer $\boldsymbol{e}$ and the radius of sampling is constrained by the covariance matrix $\boldsymbol{\Sigma} = \mathrm{diag}(\rho|v_1^+ - v_1^-|, ..., \rho|v_d^+ - v_d^-|)$, where $\rho$ is a hyperparameter, and the sampling on $i$-th dimension is half-truncated to only keep the distribution outside the hypercube as we desire to extend the boundary. With $N_e \times N_s$ auxiliary explorers in total, we query the original model and use Beam Search to select $N_e$ samples with the minimal probability of being normal as the candidate explorers for the next iteration.

**Gradient Approximation (line 7~9).** Though we have obtained $N_e$ candidate explorers in the previous step, using them directly for the next iteration does not guarantee the optimal direction of movement towards the decision boundary. To find the optimal direction, we utilize the Fast Gradient Sign Method [45] that employs gradient ascent to find the direction of feature perturbation. However, we do not know the loss function of the original model in black-box scenarios. To deal with it, given a selected auxiliary explorer $\hat{\boldsymbol{e}}$ that is sampled around an initial explorer $\boldsymbol{e}$ on the $i$-th dimension hyperplane, we approximate the $i$-th dimension of the model gradient (i.e., the partial derivative) by the slope of a linear model across the two data points, and use the midpoint with its $i$-th dimension minus the approximation as the new explorer for the next iteration:

$$e_{i,next} = \frac{e_i + \hat{e}_i}{2} - \eta \cdot sign(\nabla_i), \text{where } \nabla_i = \frac{\partial f(\boldsymbol{x})}{\partial x_i} \approx \frac{f(\boldsymbol{e}) - f(\hat{\boldsymbol{e}})}{e_i - \hat{e}_i}, \tag{8}$$

$sign(\cdot)$ is the sign function, and $\eta$ is a hyperparameter to control the stride of one iteration. The iteration stops when i) an auxiliary explorer $\hat{\boldsymbol{e}}_{ext}$ that satisfies $f(\hat{\boldsymbol{e}}_{ext}) < \varphi$ is found, or ii) it reaches the maximum number of iterations.

**Rule Acquisition (line 12).** If the iteration stops due to the first condition, we produce a boundary constraint for each dimension using the coordinate of $\hat{\boldsymbol{e}}_{ext}$ that extends the boundary of the hypercube, i.e., $c_i = (x_i \odot \hat{e}_{ext,i})$, where $\odot$ is "$\leq$" if $\hat{e}_{ext,i}$ is greater than $v_i^+$, or "$>$" if $\hat{e}_{ext,i}$ is less than $v_i^-$. If the iteration stops due to the second condition, it means the algorithm encounters difficulties in moving towards the decision boundary by perturbing this feature dimension. We calculate the difference between the model prediction of the last auxiliary explorer and that of the initial explorers on the hyperplane. If the difference is smaller than a threshold $\delta$, we decide that this feature dimension is a *contour line*, i.e., it has no significant correlation with the model prediction. In this case, we do not produce any constraints for this dimension. If the difference is greater than the threshold, we produce constraints in the same way as those produced under the first condition. The final boundary inference rule is the disjunction of the hypercube and the constraints on each dimension.

## 6 Evaluation

### 6.1 Experimental Setup

**Black-box Models and Datasets.** We use four different types of unsupervised anomaly detection models widely used in security applications as the original black-box models, including autoencoder (AE, used by [1]), variational autoencoder (VAE, used by [46]), one-class SVM (OCSVM, used by

---

**Algorithm 1:** Compositional Boundary Exploration

---

**Input:** Data falling into the $k$-th leaf node $\boldsymbol{X}_k$, anomaly detector $f$ and its threshold $\varphi$
**Output:** Boundary inference rule $C_k$ on this leaf node such that $C_k$ encapsulates normality

**1** $H_k \leftarrow \text{MinimalHypercube}(X_k)$;
**2** **for** $i$-th dimension **in** $\boldsymbol{X}_k$ **do**
**3**      $\boldsymbol{e}^{(1)}, ..., \boldsymbol{e}^{(N_e)} \leftarrow \text{IntialExplorer}(H_k)$ on $i$-th dimension;
**4**      **while** True **do**
**5**          $\hat{\boldsymbol{e}}^{(1)}, ..., \hat{\boldsymbol{e}}^{(N_s)} \leftarrow \text{AuxiliaryExplorer}(\boldsymbol{e})$ for each initial explorer $\boldsymbol{e}$;
**6**          Beam Search for $N_e$ candidate explorers from $N_e \times N_s$ auxiliary explorers that have the minimal probability of being normal judged by $f$ and $\varphi$;
**7**          $\boldsymbol{e} \leftarrow \text{GradientApprox}(\hat{\boldsymbol{e}})$ for each candidate explorer selected from auxiliary explorers;
**8**          **if** ending condition satisfied **then**
**9**              $c_i \leftarrow (x_i \odot \hat{e}_i)$ and **break**;
**10**      **end while**
**11** **end for**
**12** **return** $C_k^E = H_k \vee (c_1 \wedge c_2 \wedge ... \wedge c_d)$;

---

[47]) and Isolation Forest (iForest, used by [48]). We employ three benchmark datasets for network intrusion detection in the experiment, including CIC-IDS2017, CSE-CIC-IDS2018 [49] and TON-IoT [50]. The representation of these datasets is tabular data, where each row is a network flow record and each column is a statistical attribute, such as the mean of packet sizes and the inter-arrival time. The datasets are randomly split by the ratio of 6:2:2 for training, validation and testing. We use only normal data to train the anomaly detection models and calibrate their hyperparameters. The description of the datasets and the AUC score of the models on the datasets are shown in Table 1.

**Baselines.** We employ five prior explanation methods as baselines: 1) We use [38] that extracts rules from unsupervised anomaly detection (UAD); 2) For other global methods, we use the estimated greedy decision tree (EGDT) proposed by [15], and Trustee [16] that specifically explains security applications; 3) We also consider one method LIME [6] that can use a Submodular Pick algorithm to aggregate local explanations into global explanations, and a knowledge distillation (KD) method [43] that globally converts a black-box model to a self-explained decision tree. These methods, like ours, can only access normal data to extract explanations. More details about baselines are in the appendix.

**Metrics.** We refer to the metrics in [18] to evaluate the rule extraction. Due to limited space, we demonstrate the following four metrics in this section and present other results in the appendix: 1) Fidelity (FD), i.e., the ratio of input samples on which the predictions of original models and surrogate models agree over the total samples, which indicates the extent to which humans can trust the explanations; 2) Robustness (RB), i.e, the persistence of the surrogate model to withstand small perturbations of the input that do not change the prediction of the original model; 3) True positive rate (TPR) and true negative rate (TNR), suggesting whether the detection capability meets the need of online defense and whether the extracted rules generate noticeable false alarms that cause "alert fatigue" [51] in highly unbalanced scenarios of most security applications, respectively.

## 6.2 Quality of Rule Extraction

We extract rules from the four unsupervised anomaly detection models using the five baseline methods and our method, and test the performance of the extracted rules. The results on the three datasets are in Table 2. We find that our method achieves the highest fidelity on all the detection models and datasets, and half of the scores are even over 0.99. It shows our method can precisely match the predictions of the black box models, which ensures the correctness of its global interpretation.

Table 1: Summary of datasets for network intrusion detection and AUC of trained models.

| No. | Dataset | #Classes | #Features | #Normal | #Attack | AE | VAE | OCSVM | iForest |
|-----|---------|----------|-----------|---------|---------|-----|-----|-------|---------|
| 1 | CIC-IDS2017 | 6 attacks + 1 normal | 80 | 687,565 | 288,404 | 0.9921 | 0.9901 | 0.9967 | 0.9879 |
| 2 | CSE-CIC-IDS2018 | 14 attacks + 1 normal | 80 | 693,004 | 202,556 | 0.9906 | 0.9767 | 0.9901 | 0.9734 |
| 3 | TON-IoT | 9 attacks + 1 normal | 30 | 309,086 | 893,006 | 0.9998 | 0.9998 | 0.9993 | 0.9877 |

Table 2: Performance of rule extraction on different datasets.

CIC-IDS2017 dataset

| Method | AE | | | | VAE | | | | OCSVM | | | | iForest | | | |
|---|---|---|---|---|---|---|---|---|---|---|---|---|---|---|---|---|
| | FD | RB | TPR | TNR | FD | RB | TPR | TNR | FD | RB | TPR | TNR | FD | RB | TPR | TNR |
| UAD | 0.1325 | 0.4991 | 0.0003 | 0.9792 | 0.1438 | 0.4839 | 0.022 | **0.9988** | 0.0725 | 0.5000 | 0.00 | 1.00 | 0.1262 | 0.5000 | 0.0 | **1.00** |
| EGDT | 0.533 | **1.00** | 0.4354 | 0.9947 | 0.1437 | **1.00** | 0.022 | 0.9961 | 0.9189 | 0.9994 | 0.9306 | 0.838 | 0.9729 | 0.9996 | 0.9417 | 0.9189 |
| Trustee | 0.4871 | 0.6412 | 0.3844 | 0.9981 | 0.1552 | 0.9857 | 0.0152 | **0.9988** | 0.539 | 0.6108 | **1.00** | **1.00** | 0.4543 | 0.5801 | 0.9795 | 0.4486 |
| LIME | 0.6918 | 0.9999 | 0.7889 | 0.0014 | 0.8232 | **1.00** | 0.9329 | 0.001 | 0.068 | 0.9999 | 0.0777 | 0.0241 | 0.8910 | **0.9998** | 0.8246 | 0.9913 |
| KD | 0.5776 | 0.9989 | 0.4792 | **0.9998** | 0.2010 | 0.9817 | 0.1016 | 0.9993 | 0.3620 | **1.00** | 0.3102 | 0.9995 | 0.1262 | 0.7016 | 0.00 | **1.00** |
| Ours | **0.9835** | **1.00** | **0.9457** | 0.9915 | **0.9620** | 0.9993 | **0.9610** | 0.9944 | **0.9275** | **1.00** | **1.00** | **1.00** | **1.00** | 0.9949 | **0.9968** | 0.9843 |

CSE-CIC-IDS2018 dataset

| Method | AE | | | | VAE | | | | OCSVM | | | | iForest | | | |
|---|---|---|---|---|---|---|---|---|---|---|---|---|---|---|---|---|
| | FD | RB | TPR | TNR | FD | RB | TPR | TNR | FD | RB | TPR | TNR | FD | RB | TPR | TNR |
| UAD | 0.3796 | 0.3077 | 0.0004 | 0.7418 | 0.2697 | 0.2930 | 0.1490 | 0.4857 | 0.6051 | 0.3069 | 0.3004 | 0.9876 | 0.6811 | 0.4035 | 0.3539 | 0.9724 |
| EGDT | 0.5821 | **1.00** | 0.1432 | 0.9801 | 0.2197 | 0.9989 | 0.2308 | 0.9554 | 0.5106 | **1.00** | **1.00** | 0.9546 | 0.9546 | 0.7813 | 0.9888 | 0.8971 |
| Trustee | 0.5157 | 0.9006 | 0.1901 | 0.9857 | 0.3642 | 0.9752 | 0.0124 | 0.9636 | 0.3616 | 0.5955 | **1.00** | **1.00** | 0.4241 | 0.4700 | 0.9641 | 0.5162 |
| LIME | 0.5838 | 0.9997 | 0.7681 | 0.0255 | 0.6814 | **1.00** | 0.9402 | 0.0213 | 0.0560 | 0.9999 | 0.9999 | 0.0186 | 0.8903 | **1.00** | 0.9884 | 0.8745 |
| KD | 0.5074 | 0.9999 | 0.3562 | **0.9979** | 0.4234 | 0.9989 | 0.1086 | 0.9925 | 0.3180 | 0.9967 | 0.4308 | 0.1510 | 0.3596 | 0.6834 | 0.0000 | **1.00** |
| Ours | **0.9954** | 0.9997 | **0.9998** | 0.9774 | **0.8962** | 0.9985 | **0.9997** | 0.8268 | **0.9929** | 0.9997 | 0.9983 | 0.9753 | **0.9947** | 0.9291 | **0.9988** | 0.9583 |

TON-IoT dataset

| Method | AE | | | | VAE | | | | OCSVM | | | | iForest | | | |
|---|---|---|---|---|---|---|---|---|---|---|---|---|---|---|---|---|
| | FD | RB | TPR | TNR | FD | RB | TPR | TNR | FD | RB | TPR | TNR | FD | RB | TPR | TNR |
| UAD | 0.1499 | 0.015 | 0.0258 | 0.908 | 0.2157 | 0.4010 | 0.1863 | 0.7787 | 0.0489 | 0.5000 | 0.00 | **1.00** | 0.0674 | 0.5000 | 0.00 | **1.00** |
| EGDT | 0.9750 | **1.00** | 0.9739 | 0.9943 | 0.7660 | **1.00** | 0.7538 | 0.9948 | 0.8139 | 0.9997 | 0.8051 | 0.9759 | 0.6345 | 0.9226 | 0.6247 | 0.9475 |
| Trustee | 0.4774 | 0.5722 | 0.4502 | 0.9971 | 0.3807 | 0.6689 | 0.3484 | 0.9975 | 0.7942 | 0.8430 | **1.00** | **1.00** | 0.7476 | 0.8145 | 0.9824 | 0.1943 |
| LIME | 0.6971 | 0.9999 | 0.7939 | 0.0027 | 0.8289 | **1.00** | 0.9379 | 0.0015 | 0.0687 | 0.9999 | 0.0787 | 0.0231 | 0.8963 | **0.9998** | 0.8296 | 0.9918 |
| KD | 0.0821 | **1.00** | 0.0341 | **0.9987** | 0.0591 | 0.9997 | 0.0099 | **0.9980** | 0.0494 | **1.00** | 0.0005 | 0.9994 | 0.0674 | 0.9955 | 0.00 | **1.00** |
| Ours | **0.9996** | **1.00** | **1.00** | 0.9845 | **0.9995** | **1.00** | **1.00** | 0.9831 | **0.9511** | **1.00** | **1.00** | 0.9881 | **1.00** | 0.9890 | **1.00** | 0.9715 |

Table 3: Fidelity of extracted rules under varying percentages of noisy training data.

| Percentage | Random Noise | | | | Mislabeled Noise | | | |
|---|---|---|---|---|---|---|---|---|
| | AE | VAE | OCSVM | iForest | AE | VAE | OCSVM | iForest |
| 0% | 0.9829 | 0.9814 | 0.9729 | 0.9876 | 0.9997 | 0.9977 | 0.9975 | 0.9984 |
| 1% | 0.9829 | 0.9824 | 0.9148 | 0.9940 | 0.9991 | 0.9992 | 0.9952 | 0.9953 |
| 3% | 0.9876 | 0.9873 | 0.8960 | 0.9920 | 0.9991 | 0.9992 | 0.9952 | 0.9953 |
| 5% | 0.9855 | 0.9675 | 0.9511 | 0.7732 | 0.9914 | 0.9996 | 0.9992 | 0.9966 |
| 10% | 0.9739 | 0.9881 | 0.9600 | 0.5148 | 0.9987 | 0.9983 | 0.9996 | 0.9978 |

Moreover, our method achieves the highest TPR on all the detection models and datasets; specifically, the TPR is equal to 1.00 for all the detection models on TON-IoT dataset. This result suggests that our rules can accurately detect various anomalous data, making it possible to realize online deployment and defense using these rules. Our method also reaches a high level of robustness (minimum 0.9890, maximum 1.00) and true negative rate (minimum 0.9715, maximum 1.00). Therefore, it is concluded that our method can obtain rules of high quality from different black-box unsupervised anomaly detection models using only unlabeled one-class data.

Considering that obtaining a "clean" training set requires huge manual effort in reality [52], we also assess the efficacy of our method under varying percentages of "noisy" data. We evaluate the fidelity of extracted rules using two approaches for the injection of noisy data: 1) random noise; 2) mislabeled data from other classes, i.e., attack data. The results are shown in Table 3. We find that the impact of the noisy data proportion is not significant: 36 of 40 fidelity scores in the table preserve over 0.95, and the variation of fidelity scores is not obvious with the increase of noisy data for most of the models. This shows that our rule extraction method can retain similar performance to the black-box model that it extracts from. Nonetheless, the results of iForest also reveal that a sufficiently large proportion of noisy data may cause a certain negative impact on the rule extraction for certain models.

## 6.3 Understanding Model Decisions

To demonstrate that the rules obtained by our method are in line with human understanding, we use the OCSVM as an example of black-box models to exhibit several explanations. We extract rules from the well-train model and use the rules to predict three typical types of attack data, including Distributed Denial-of-Service (DDoS) attacks, scanning attacks, SQL injection, and backdoor attacks.

Table 4: Examples of explanation on four types of attacks.

| Attack | Rules of Normality | Attack Value | Feature Meaning | Human Understanding |
|--------|-------------------|--------------|-----------------|---------------------|
| DDoS | $ps\_mean > 101.68$
$iat\_mean > 0.063$
$dur > 12.61$ | 57.33
0.00063
0.00126 | Mean of IP packet sizes
Mean of packet inter-arrival time
Duration of a connection | DDoS attacks use packets of small sizes to achieve
asymmetric resource consumption on the victim side,
and send packets at a high rate to flood the victim. |
| Scanning | $count > 120$
$ps\_var > 2355.20$ | 1
0.0 | IP packet count per connection
Variance of IP packet sizes | Scanning attacks send a constant probe packet to a port,
and the victim will not reply if the port is closed. |
| SQL Injection | $ps\_bwd\_mean \leq 415.58$
$dur > 1.64$ | 435.80
0.37 | Mean of backward IP packet sizes
Duration of a connection | Unauthorized access to additional data from websites,
usually establish short connections for one attack. |
| Backdoor | $ps\_max > 275.28$
$ps\_min > 49.41$ | 48.0
40.0 | Maximum of IP packet sizes
Minimum of IP packet sizes | It persists in compromised hosts and sends stealthy
keep-alive packets with no payload (thus very small). |

Table 4 shows some features of the rules extracted from normal data that cannot be matched by the attack data, and exhibits how humans can interpret the model decisions[2]. For example, the data of DDoS attacks cannot match the rules of three feature dimensions, including mean of packet sizes, mean of packet inter-arrival time, and duration of a connection. It can be observed that the feature values of attacks are markedly lower than the bound of the rules. Such results are easy to interpret. Because the purpose of DDoS attacks is to overwhelm the resources of a victim, an attacker will realize asymmetric resource consumption between the victim and himself (i.e., using small packets), send packets at an extremely high rate (i.e., low inter-arrival time), and establish as many useless connections as possible (i.e., short duration of connections). These explanations are in line with how humans recognize the attack data. Hence, we can draw a conclusion that our method is able to provide precise insights into black-box anomaly detection models in a human-understandable way.

## 6.4 Ablation Study

To evaluate the contribution of each component in our method, including the IC-Tree and the CBE algorithm, we conduct an ablation experiment by 1) replacing the IC-Tree with a clustering algorithm K-Means, 2) using only the CBE algorithm, and 3) replacing the CBE algorithm with directly using hypercubes as rules. In Table 5, we find that our method (IC-Tree + CBE) outperforms others in terms of fidelity on both datasets. Though using the K-Means can reach similar results, it cannot be expressed by axis-aligned rules with high interpretability and deployability as the IC-Tree can achieve. In summary, both components are helpful for the quality of rule extraction.

Table 5: Ablation study on the components of our method.

| Method | CIC-IDS2017 | | TON-IoT | |
|--------|-------------|------------|---------|------------|
| | FD | Comparison | FD | Comparison |
| IC-Tree + CBE (our method) | 0.9856 | - | 0.9840 | - |
| K-Means (k=10) + CBE | 0.9731 | 1.268%↓ | 0.9802 | 0.386%↓ |
| K-Means (k=5) + CBE | 0.9735 | 1.228%↓ | 0.9793 | 0.386%↓ |
| Only CBE | 0.9735 | 1.228%↓ | 0.9784 | 0.478%↓ |
| IC-Tree + Hypercube | 0.9652 | 2.069%↓ | 0.0647 | 93.43%↓ |

## 6.5 Computational Cost and Complexity

We also evaluate the computational cost of our method with respect to training and prediction. Since CIC-IDS2017 dataset has 80 features in total, we train the model using the first 20, 40, 60, and 80 features of 4000 samples to investigate the influence of feature sizes. The results are shown in Table 6, which demonstrate the average training and prediction time of our method. It can be seen that the training time is around 1 minute, which is acceptable and practical for large-scale training. Besides, the training time increases basically linearly with the increase of feature sizes. This is because our method adopts a feature-by-feature strategy to explore the decision boundary of the model. For prediction time, our method is highly efficient, which only costs microsecond-level overhead for one inference. It shows that as a rule-based approach, our method can achieve real-time execution for online use. Note that the runtime is measured purely based on Python. In practice, the prediction time of our method can be even less with more efficient code implementation.

---

[2]Note that the "human understanding" was derived from the knowledge of the authors, and hence may be subjective and not reflect the wide population of security experts. We give more clarification of obtaining the content of Table 4 in the appendix, as well as potential reasons for the disagreement between humans and models.

Table 6: Average training and prediction time per sample for different feature sizes.

| Feature Size | Training Time (ms) | Prediction Time (ms) |
|---|---|---|
| 20 | $5.40 \pm 5.50 \times 10^{-4}$ | $5.48 \times 10^{-3} \pm 2.51 \times 10^{-9}$ |
| 40 | $15.5 \pm 6.80 \times 10^{-2}$ | $5.52 \times 10^{-3} \pm 2.34 \times 10^{-9}$ |
| 60 | $14.7 \pm 8.75 \times 10^{-5}$ | $6.99 \times 10^{-3} \pm 3.56 \times 10^{-8}$ |
| 80 | $30.7 \pm 3.08 \times 10^{-1}$ | $6.91 \times 10^{-3} \pm 9.00 \times 10^{-8}$ |

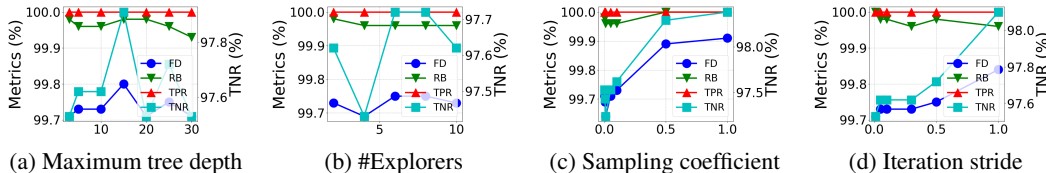

(a) Maximum tree depth    (b) #Explorers    (c) Sampling coefficient    (d) Iteration stride

Figure 2: Sensitivity experiments of hyperparameters.

We also theoretically analyze the time complexity of our algorithms. For training, the complexity of the IC-Tree is identical to a CART: $O(d \cdot n \log n)$, where $d$ is the feature size and $n$ is the sample number; the complexity of the CBE algorithm is $O(K \cdot d \cdot N_e \cdot N_s)$, where $K$ is the number of leaf nodes of the IC-Tree, and $N_e$ and $N_s$ are the number of initial explorers and auxiliary explorers. Therefore, the training time is theoretically linear to the feature size, which is in line with the empirical results. For execution, the time complexity is $O(|C| \cdot d)$, where $|C|$ is the number of extracted rules.

## 6.6 Hyperparameter

We perform a sensitivity analysis of several hyperparameters on their influence on the rule extraction. We present four major hyperparameters in Figure 2, including the maximum depth $\tau$ of an IC-Tree, $N_e$ number of explorers, the coefficient $\rho$ of sampling, and the factor $\eta$ that controls the stride of an iteration. Due to limited space, the analysis of other hyperparameters is placed in the appendix.

**Maximum tree depth.** A deeper IC-Tree has more leaf nodes, and can accordingly decompose more distributions that ease the difficulty of rule extraction. Meanwhile, excessively fine-grained splitting might cause overfitting. We find that $\tau = 15$ achieves the best performance.

**Number of Explorers.** It is essentially the number of selected nodes per iteration in Beam Search, which considers multiple local optima to improve greedy algorithms. But selecting too many nodes may also include more redundancy. Figure 2b shows that a value between 6 and 8 is recommended.

**Coefficient of sampling.** Figure 2c shows that a higher value of the hyperparameter achieves better results. A large coefficient decides a large radius of sampling from a multivariant Gaussian distribution, which helps the CBE algorithm quickly find the decision boundary of the original model.

**Factor of iteration stride.** In Figure 2d, we find that a larger factor $\eta$ can obtain rules of higher quality. As it decides the stride of finding the explorers for the next iteration, a higher value of the hyperparameter might help the convergence of the iteration process.

## 7 Conclusion and Future Work

This paper proposes a novel method to globally interpret black-box unsupervised anomaly detection, in which we introduce two types of rules and corresponding algorithms to extract these rules. The evaluation shows that our method outperforms prior work in terms of various metrics, boosting user confidence in complex models and facilitating their integration into high-stake security applications.

There are numerous meaningful directions for future work. First, new algorithms can be developed based on model interpretation to troubleshoot illogical inference processes in black-box models (e.g., false positives) and realize the automation of precisely fixing errors. Moreover, researchers can investigate the integration of rule extraction with high-performance rule-based defense systems, such as P4 [13], to implement a more efficient security system. In addition, since the proposed method can be generic, the transfer to other areas of industry that also demand interpretable anomaly detection, such as health, manufacturing and criminal investigation, can be further explored.

## Acknowledgments and Disclosure of Funding

This work is supported by the National Key Research and Development Program of China under grant No. 2022YFB3105000, the Major Key Project of PCL under grant No. PCL2023AS5-1, the Shenzhen Key Lab of Software Defined Networking under grant No. ZDSYS20140509172959989, and the research fund of Tsinghua University - Tencent Joint Laboratory for Internet Innovation Technology.

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

# A  Appendix

## A.1  Proof of Proposition 1

**Our Goal.** We expect the extracted rules to have a high fidelity to the original model, that is, a similar coverage of normal data (i.e., true negative rate), and a similar detection rate of anomalies (i.e., true positive rate). To this end, we formulate our objective as follows:

$$\arg\min_{\mathcal{C}} \mathcal{L}_{\mathcal{X}\sim\mathcal{D}}(\mathcal{C}, f, \varphi) + \mathcal{L}_{\mathcal{X}\not\sim\mathcal{D}}(\mathcal{C}, f, \varphi). \tag{9}$$

**Proposition 1.** *The original objective can be estimated by finding the union of the conjunction of distribution decomposition rules and boundary inference rules for each compositional distribution:*

$$\bigcup_{k=1}^{K} \arg\min_{C_k} \mathcal{L}_{\mathcal{X}\sim\mathcal{D}_k}(C_k, f, \varphi) + \mathcal{L}_{\mathcal{X}\not\sim\mathcal{D}_k}(C_k, f, \varphi), \text{where } C_k = C_k^I \wedge C_k^E. \tag{10}$$

We give the following lemma equivalent to Proposition 1 and prove it:

**Lemma 1.1.** *If the distribution decomposition rules and the inference boundary rules that minimize the loss on each of the compositional distributions are found, the sum of the minimum losses on each of the compositional distributions can estimate the minimum loss on the overall distribution with a significantly small error $\psi$, i.e.,*

$$\min \mathcal{L}_{\mathcal{X}\sim\mathcal{D}}(\mathcal{C}, f, \varphi) + \mathcal{L}_{\mathcal{X}\not\sim\mathcal{D}}(\mathcal{C}, f, \varphi)$$

$$= \sum_{k=1}^{K} \min(\mathcal{L}_{\mathcal{X}\sim\mathcal{D}_k}(C_k, f, \varphi) + \mathcal{L}_{\mathcal{X}\not\sim\mathcal{D}_k}(C_k, f, \varphi)) + \psi, \tag{11}$$

$$\text{where } C_k = C_k^I \wedge C_k^E, \psi \geq 0.$$

**Proof.** The sum of the minimum losses on each of the compositional distributions is calculated by an iteratively cumulative process. Let $L_j$ be the sum of the minimum losses on each of the compositional distributions at the $j$-th iteration, i.e.,

$$L_j = \sum_{k=1}^{j} \min(\mathcal{L}_{\mathcal{X}\sim\mathcal{D}_k}(C_k, f, \varphi) + \mathcal{L}_{\mathcal{X}\not\sim\mathcal{D}_k}(C_k, f, \varphi)).$$

Let $\mathcal{X} \sim \bigcup_{k=1}^{j} \mathcal{D}_k$ represent a variable belonging to any of the compositional distributions $\mathcal{D}_1, ..., \mathcal{D}_j$. We prove the Loop Invariant of $L_j$ during the iteration, which always satisfies:

$$L_j = \min(\mathcal{L}_{\mathcal{X}\sim\bigcup_{k=1}^{j}\mathcal{D}_k}(\bigcup_{k=1}^{j} C_k, f, \varphi) + \mathcal{L}_{\mathcal{X}\not\sim\bigcup_{k=1}^{j}\mathcal{D}_k}(\bigcup_{k=1}^{j} C_k, f, \varphi)) + \psi. \tag{12}$$

1) For the first iteration, the equation $L_1 = \min \mathcal{L}_{\mathcal{X}\sim\mathcal{D}_1}(C_1, f, \varphi) + \mathcal{L}_{\mathcal{X}\not\sim\mathcal{D}_1}(C_1, f, \varphi) + \psi$ obviously holds where $\psi = 0$.

2) Suppose the Equation (12) holds at the $j$-th iteration. For the $(j+1)$-th iteration, we have the following derivations:

$$L_{j+1} = \sum_{k=1}^{j+1} \min(\mathcal{L}_{\mathcal{X}\sim\mathcal{D}_k}(C_k, f, \varphi) + \mathcal{L}_{\mathcal{X}\not\sim\mathcal{D}_k}(C_k, f, \varphi))$$

$$= \min(\mathcal{L}_{\mathcal{X}\sim\bigcup_{k=1}^{j}\mathcal{D}_k}(\bigcup_{k=1}^{j} C_k, f, \varphi) + \mathcal{L}_{\mathcal{X}\not\sim\bigcup_{k=1}^{j}\mathcal{D}_k}(\bigcup_{k=1}^{j} C_k, f, \varphi)) + \psi$$

$$+ \min(\mathcal{L}_{\mathcal{X}\sim\mathcal{D}_{j+1}}(C_{j+1}, f, \varphi) + \mathcal{L}_{\mathcal{X}\not\sim\mathcal{D}_{j+1}}(C_{j+1}, f, \varphi))$$

$$= \min(\mathcal{L}_{\mathcal{X}\sim\bigcup_{k=1}^{j+1}\mathcal{D}_k}(\bigcup_{k=1}^{j+1} C_k, f, \varphi) + \mathcal{L}_{\mathcal{X}\not\sim\bigcup_{k=1}^{j+1}\mathcal{D}_k}(\bigcup_{k=1}^{j+1} C_k, f, \varphi)) + \psi$$

$$+ \mathcal{L}_{\mathcal{X}\sim\bigcup_{k=1}^{j}\mathcal{D}_k\cap\mathcal{D}_{j+1}}(\bigcup_{k=1}^{j+1} C_k, f, \varphi) + \mathcal{L}_{\mathcal{X}\not\sim\bigcup_{k=1}^{j}\mathcal{D}_k\cap\mathcal{D}_{j+1}}(\bigcup_{k=1}^{j+1} C_k, f, \varphi),$$

where $\bigcup_{k=1}^{j+1} \mathcal{D}_k \cap \mathcal{D}_{j+1}$ represents the overlap area between the conjunction of the compositional distributions $\bigcup_{k=1}^{j+1} \mathcal{D}_k$ and the $(j+1)$-th compositional distribution $\mathcal{D}_{j+1}$. Recall the definition of the compositional distributions that a data sample belonging to one compositional distribution has a significantly small probability of belonging to other compositional distributions, meaning that the overlap area between the compositional distributions is significantly small. Therefore, the loss with respect to the overlap area is also significantly small, given the data samples belonging to the area are significantly rare. Let

$$\psi = \psi + \mathcal{L}_{\mathcal{X} \sim \bigcup_{k=1}^{j} \mathcal{D}_k \cap \mathcal{D}_{j+1}}(\bigcup_{k=1}^{j+1} C_k, f, \varphi) + \mathcal{L}_{\mathcal{X} \nsim \bigcup_{k=1}^{j} \mathcal{D}_k \cap \mathcal{D}_{j+1}}(\bigcup_{k=1}^{j+1} C_k, f, \varphi),$$

and we can get the final result of $L_{j+1}$:

$$L_{j+1} = \min(\mathcal{L}_{\mathcal{X} \sim \bigcup_{k=1}^{j+1} \mathcal{D}_k}(\bigcup_{k=1}^{j+1} C_k, f, \varphi) + \mathcal{L}_{\mathcal{X} \nsim \bigcup_{k=1}^{j+1} \mathcal{D}_k}(\bigcup_{k=1}^{j+1} C_k, f, \varphi)) + \psi,$$

which proves the loop invariant in Equation (12). At the last iteration when $j = K$, as the overall distribution is equal to the conjunction of the compositional distributions, i.e., $\mathcal{D} = \bigcup_{k=1}^{K} \mathcal{D}_k$, we prove Equation (11) holds and Lemma 1.1 is correct.

## A.2 Implementation of Our Method

Our implementation is primarily based on PyTorch (version 1.12.1) for the deep learning models, such as AE and VAE. Additionally, for data preprocessing, feature engineering, and model evaluation, we employ the versatile machine learning library scikit-learn (version 1.1.3). Python (version 3.9.15) serves as the programming language for our implementation, providing a rich ecosystem of libraries and tools for data manipulation and experiment orchestration.

Our experiments were conducted on a server equipped with the Intel(R) Xeon(R) Gold 5218 CPU @ 2.30GHz (128GB RAM) and the GeForce RTX 2080 Super (8GB VRAM). Note that GPU is only used for the training of some DL-based anomaly detectors (e.g., AE, VAE), and our rule extraction method only requires the use of CPU.

## A.3 Implementation of Baselines

In this section, we delve into the details of the baseline methods used for evaluation in our experiments, focusing on their implementation in the context of globally explaining unsupervised anomaly detection. The five baseline methods include Rule extraction from UAD, Estimated Greedy Decision Tree (EGDT), Trustee, Local Interpretable Model-agnostic Explanations (LIME), and a Knowledge Distillation (KD) method.

### A.3.1 Rule Extraction from UAD

The Rule Extraction from UAD method was initially designed for one-class support vector machines (OCSVM) but is applicable to various types of unsupervised anomaly detection models. It uses feature scaling, anomaly detection, and clustering to produce rules from the input parameters $X, ln, lc, d, m, t$, where $X$ refers to the input data frame with the features, $ln$ is a list with the numerical columns, $lc$ is a list with the categorical columns, $d$ is a dictionary with the hyperparameters for OCSVM (kernel type, upper bound on the fraction of training errors and a lower bound of the fraction of support vectors, $\nu$, and the kernel coefficient, $\gamma$). $m$ is a variable that defines the type of cluster algorithm, and $t$ is a variable that specifies the type of approximation for obtaining the hypercubes.

The approach for obtaining rules depends on the type of cluster algorithm used and can involve iterative clustering and splitting of data points. The choice of kernel type in the OCSVM model can influence the rules obtained, with RBF kernel tending to enclose inliers and leave outliers outside, eventually forming hypercubes to represent the rules.

### A.3.2 Estimated Greedy Decision Tree (EGDT)

The EGDT method constructs a decision tree that approximates the black-box model. The algorithm generates new training data by actively sampling new inputs and labeling them with a black-box

model. Based on these training data, EGDT constructs the decision tree $T^*$ of size $k$ similar to CART, in a greedy manner and is pruned to improve interoperability. The algorithm takes into account the distribution of points that are routed to each leaf node in the decision tree to ensure that the label assigned to each leaf node is accurate.

### A.3.3 Trustee

The model-agnostic DT extraction method Trustee, specifically focuses on interpreting security applications. The core idea is to construct an interpretable decision tree by minimizing the difference between the black-box model and the surrogate model. The algorithm creates several high-fidelity decision trees by executing an outer loop $S$ times and an inner loop $N$ times. In an unsupervised setting, we treat the predictions of the black-box model as pseudo-labels and employ the same process as for the supervised case.

### A.3.4 Local Interpretable Model-agnostic Explanations (LIME)

By learning an interpretable model locally around the prediction, LIME explains the predictions of any classifier in an interpretable and faithful manner. LIME uses locally weighted square loss and an exponential kernel to approximate the black-box model. LIME trains a sparse linear model on the local dataset to explain the prediction. The feature importance scores assist in identifying the most important features that contribute to the prediction, which can be used to gain insights into the model and improve LIME's trustworthiness.

In addition, this paper proposes "SP-LIME" that selects a set of representative instances with explanations to address the "trusting the model" problem, via submodular optimization. The submodular pickup method is used to select a subset of instances that are representative of the entire dataset while reducing redundancy. This method allows for a global perspective to ascertain trust in the model, in addition to explaining individual predictions. LIME can therefore be transformed from a local method to a global method using the submodular pickup method, which makes it more useful in real scenarios.

The implementation of LIME in unsupervised settings is fairly straightforward. For a given input instance, we create a set of perturbations, predict their labels using the black-box model, and then fit an interpretable model (usually a linear model) on this newly created dataset to approximate the local decision boundary.

### A.3.5 Knowledge Distillation (KD) Method

In order to achieve interpretability and effectiveness simultaneously, the KD method proposes a knowledge distillation-based decision tree extension, called rectified decision trees (ReDT), and tries to transfer the knowledge from a complex model (the teacher) to a simpler one (the student). ReDT extends the splitting criteria and the ending condition of the standard decision trees to allow training with soft labels distilled from a well-trained teacher model while preserving the deterministic splitting paths. They recommend a jackknife-based distillation to obtain soft labels, which does not require backpropagation for the student model, and analyze the effectiveness of adopting soft labels instead of hard ones empirically and theoretically. In an unsupervised scenario, we employ the anomaly scores produced by the teacher model as pseudo-labels. The student model is then trained to imitate these scores. This method's limitation lies in its strong reliance on the correctness of the teacher model's scores.

We need to note that these unsupervised adaptations of the baselines essentially introduce a supervised flavor to the learning process, with the black-box model's predictions serving as the pseudo-labels. It is an important observation that these methods do not inherently work in an unsupervised manner and their application in such settings might be unreliable depending on the accuracy of the black-box model's predictions. On the other hand, our proposed method is fundamentally designed for unsupervised settings, making it a more trustworthy choice for real-world applications.

It is also worth mentioning that, except for **LIME**[3] and **Trestee**[4] which use open-source code, all other baseline methods were implemented in Python using the scikit-learn library, with default

---

[3] https://github.com/marcotcr/lime
[4] https://github.com/TrusteeML/trustee

parameters unless specified otherwise. Hyperparameters were set based on the initial grid search and then manually fine-tuned.

## A.4 More Details in Experiments

### A.4.1 More Metrics

We also examine other metrics in addition to the metrics given in the main article.

- **Precision (PR)** measures the proportion of correctly predicted positive instances out of the total instances anticipated as positive. It indicates the model's ability to avoid false positives. A lower rate of false positives is indicated by precision values that are higher.

- **Correctness (CR)** evaluates the precision of the explanations provided by the IC-Tree method with respect to the underlying model. This can be quantified using the Jaccard similarity index, defined as $Cr = \frac{r}{N}$, where $r$ is the number of correct outputs by the model.

- **Recall (RC)** also known as sensitivity or true positive rate, quantifies the proportion of correctly predicted positive instances out of all actual positive instances. It indicates the model's ability to identify all positive instances without missing any. Higher recall values indicate a lower rate of false negatives. It can be quantitatively assessed by the ratio $RC = \frac{TP}{TP+FN}$.

- **F1 score (F1)** is the harmonic mean of precision and recall combined into a single statistic, which provides a balanced measure of both precision and recall, capturing the overall performance of the model.

Besides the aforementioned metrics, in fact, we believe that the most appropriate way to assess "human's trust in anomaly detection models" should be employing a trial in which human security experts are invited to inspect the rules of interpretation, measuring the consistency of understanding. Yet this is not an easy task to conduct: to eliminate the subjective influence of each expert, enough people must be invited to participate in the trial, which can be somewhat difficult to carry out. Despite the intrinsic limitations of lacking such a metric, it is indeed a common practice to employ only those commonly used metrics (e.g., fidelity, F1 score) even in research works accepted to top venues, which may reveal an obvious gap between "research" and "practice".

The specifics of the performance metrics of our proposed method in comparison to other baseline methods, including UAD, EGDT, Trustee, LIME, and KD, are demonstrated in Table 7 and Table 8. In addition, we analyze the effects of several hyperparameters on our methodology, as shown in Table 9. The performance metrics are categorized based on different anomaly detection algorithms: Autoencoder (AE), Variational Autoencoder (VAE), One-Class SVM (OCSVM), and Isolation Forest (iForest) are shown in Table 8. Precision (PR), Correctness (CR), Recall (RC), and F1 score (F1) are evaluated in this paper. Notably, our proposed method achieves the highest performance in almost all categories, with a few significant instances where it ties with the best performer. This indicates the comprehensive superiority of our approach, as it consistently delivers exceptional performance across a wide range of models and metrics.

Table 7: Performance of rule extraction on CIC-IDS2017 dataset.

| Method | AE | | | | VAE | | | | OCSVM | | | | iForest | | | |
|---|---|---|---|---|---|---|---|---|---|---|---|---|---|---|---|---|
| | PR | CR | RC | F1 | PR | CR | RC | F1 | PR | CR | RC | F1 | PR | CR | RC | F1 |
| UAD | 0.5896 | 0.0659 | 0.1027 | 0.175 | 0.6104 | 0.0874 | 0.1126 | 0.1901 | 0.9912 | 0.067 | 0.999 | 0.9951 | 0.9993 | 0.067 | 0.9292 | 0.963 |
| EGDT | 0.5729 | 0.4729 | 0.4729 | 0.8924 | 0.5613 | 0.873 | 0.6864 | 0.5462 | 0.9904 | 0.9244 | 0.8553 | 0.9179 | 0.9942 | 0.9402 | 0.9938 | 0.994 |
| Trustee | 0.8651 | 0.4255 | 0.6482 | 0.4255 | 0.812 | 0.831 | 0.7655 | 0.828 | 0.856 | 0.5509 | 0.6664 | 0.7494 | 0.954 | 0.4842 | 0.8676 | 0.8952 |
| LIME | 0.9998 | 0.7614 | 0.7569 | 0.8616 | 0.7091 | 0.7331 | 0.8641 | 0.9003 | 0.9892 | 0.741 | 0.9263 | 0.9567 | 0.8641 | 0.8305 | 0.8305 | 0.8751 |
| KD | 0.5141 | 0.5141 | 0.6871 | 0.5835 | 0.5612 | 0.1618 | 0.4516 | 0.5623 | 0.9157 | 0.3564 | 0.3748 | 0.5319 | 0.7863 | 0.067 | 0.8465 | 0.8461 |
| Ours | 0.9994 | 0.9488 | 0.9457 | 0.9718 | 0.9633 | 0.9633 | 0.9633 | 0.9645 | 0.9351 | 0.933 | 1 | 0.9653 | 0.933 | 0.9933 | 0.9354 | 0.9841 |

This extensive evaluation underlines the robustness and effectiveness of our method compared to established baselines. The superior performance, across varying metrics and under different detection models, signifies its potential as a versatile solution for rule extraction in the TON-IoT dataset. This

Table 8: Performance of rule extraction on TON-IoT dataset.

| Method | AE | | | | VAE | | | | OCSVM | | | | iForest | | | |
|---|---|---|---|---|---|---|---|---|---|---|---|---|---|---|---|---|
| | PR | CR | RC | F1 | PR | CR | RC | F1 | PR | CR | RC | F1 | PR | CR | RC | F1 |
| UAD | 0.0058 | 0.085 | 0.1257 | 0.182 | 0.6153 | 0.216 | 0.3546 | 0.6845 | 0.9847 | 0.0501 | 0.0526 | 0.0998 | 0.9949 | 0.0501 | 0.1776 | 0.3014 |
| EGDT | 0.9749 | 0.9749 | 0.9854 | 0.9749 | 0.7659 | 0.7659 | 0.7856 | 0.8621 | 0.9979 | 0.8136 | 0.651 | 0.788 | 0.9964 | 0.6408 | 0.7977 | 0.886 |
| Trustee | 0.8475 | 0.4776 | 0.5465 | 0.4776 | 0.3809 | 0.3809 | 0.8416 | 0.3809 | 0.8482 | 0.7932 | 0.4147 | 0.557 | 0.9548 | 0.2338 | 0.6576 | 0.7525 |
| LIME | 0.7651 | 0.7664 | 0.6845 | 0.7561 | 0.9035 | 0.9053 | 0.8486 | 0.9053 | 0.7896 | 0.0749 | 0.7815 | 0.7512 | 0.7987 | 0.8354 | 0.8861 | 0.8354 |
| KD | 0.5765 | 0.0824 | 0.6548 | 0.3554 | 0.5648 | 0.0594 | 0.5486 | 0.5461 | 0.8966 | 0.0505 | 0.1004 | 0.1805 | 0.5154 | 0.0501 | 0.8456 | 0.8465 |
| Ours | 0.9992 | 0.9992 | 0.9345 | 0.9801 | 0.9992 | 0.9992 | 0.9861 | 0.9814 | 0.9345 | 0.9499 | 1 | 0.9743 | 0.9933 | 0.9933 | 0.9456 | 0.9968 |

demonstrates the practical utility of our proposed method, especially in a diverse and dynamic domain such as IoT, which could significantly benefit from such adaptable and high-performing solutions.

As for the baseline methods, while some of them can achieve acceptable fidelity for certain models, they fail to maintain such results on other models, indicating that they cannot achieve qualified model-agnostic global explanations for unsupervised models. Further, most of their recall scores cannot meet the requirement of using their rules for online defense. It is mainly because these methods either require labeled data to determine the boundary between normal and abnormal (e.g., EGDT and LIME), or need sufficient outliers in the training data (e.g., UAD and KD), which can be unavailable in many security applications. In contrast, our method eliminates these requirements by the IC-Tree and CBE algorithm that explores the decision boundary in an unsupervised manner, and meanwhile realizes a high detection rate of anomalies.

Lastly, it is worth noting that while our method exhibits superior performance, we do not imply the obsoleteness of other methods. Every method has its strengths and use cases, thus the selection of the method should always be context-dependent. Future work could involve fine-tuning our method to improve its performance further or be applied to other domains.

### A.4.2 More Clarification of Human Understanding

We present a step-by-step description of obtaining the content in Table 4 in an ideal experiment, which takes data points as input and then shows the steps that come up with the explanations:

1. For a reported anomaly $x$, we use the IC-Tree to pinpoint the rule of normality that judges $x$ as anomalous. It is realized by inputting $x$ into the tree and recursively finding the leaf node. The rule is denoted by $C_x$.
2. For each constraint of features in $C_x$, we compare the corresponding feature value of $x$ with the range of the constraint.
3. For the feature values outside the range of the constraints, the Rules of Normality, Feature Values, and Feature Meaning (i.e., the three columns in Table 4) along with the raw data sample will be sent as the explanation to the security expert for further analysis.
4. The security expert will analyze the data sample to determine the type of attack (i.e., the Attack column in Table 4) and give her/his understanding of the important attributes that make her/him identify the attack (i.e., the Human Understanding column in Table 4).
5. If there is a huge gap between the provided explanation and expert understanding, it indicates that the anomaly detector is not trustworthy. The security expert may conduct further actions to improve the detector, such as retraining, fine-tuning, or reconsidering feature selection.

Note that such a gap may not only occur when the anomaly detector makes erroneous decisions but also when the anomaly detector correctly detects an attack sample by unreasonable features. For example, the most obvious attribute of a Denial-of-Service (DoS) attack is typically its high rate of forwarding packets in order to overwhelm the victim. However, the provided explanation might suggest the IP address is an important feature for the anomaly detector to make this inference, which is quite doubtful from the perspective of a security expert. In practice, this may be due to "spurious correlation" or "shortcut learning"[5]. Specifically, for the DoS example mentioned in our paper, unreliable decisions could be made due to inappropriate testbed settings for the collection of

---

[5]D. Arp, E. Quiring, et al., "Dos and Don'ts of Machine Learning in Computer Security", in 31st USENIX Security Symposium, USENIX Security 2022.

training data, such as the DoS attack being launched from one separate host address while all other normal traffic is from other host addresses.

### A.4.3  More Hyperparameter Analysis

The robustness of our method against changes in hyperparameters also needs to be evaluated. Sensitivity analysis analysis us to identify which parameters are most influential, thereby concentrating on tuning those to achieve optimal performance. In our study, we perform a sensitivity analysis on the hyperparameters: Maximum Iteration, Auxiliary Explorer Sample Size ($N_s$), Distribution Similarity Threshold ($\epsilon$), and Significance Threshold ($\delta$)).

**Maximum Iteration** represents the maximum number of iterations that the algorithm will run. In our analysis, as shown in Table 9, this parameter's variation from 0.01 to 0.5 has a negligible effect on all the measured indicators. This displays the algorithm's resilience and robustness to changes in this hyperparameter, indicating that the model converges relatively rapidly.

**Auxiliary Explorer Sample Size** ($N_s$) represents the number of samples gathered to assist the model's exploration phase. The results indicate that our model effectively makes use of the available samples because an increase in $Ns$ has little to no effect on the majority of the metrics. However, a slight increase in fidelity, correctness, and accuracy is observed at larger sample sizes, indicating that providing more samples can marginally enhance the model's performance.

**Distribution Similarity Threshold** ($\epsilon$) dictates how similar the proposed and baseline distributions should be for the tree's branches to accept the proposed distribution. Interestingly, whereas other metrics practically remain constant as $epsilon$ goes from 10 to 150, fidelity, correctness, and accuracy show a slight uptick. This suggests that our model exhibits an adaptable behavior to changes in distribution similarity thresholds.

**Significance Threshold** ($\delta$), which determines the minimum difference that should be deemed noteworthy, also shows a negligible effect on all the performance indicators. This shows that the model is not sensitive to changes in $\delta$ and supports the robustness of the proposed approach.

One noticeable aspect across all parameters is the TPR and TNR values. The TPR is consistently at its maximum, showcasing the system's excellent ability to identify positive instances. TNR also remains high, demonstrating the system's effectiveness in accurately identifying negative instances.

This sensitivity analysis underlines the robustness and adaptability of our proposed method. Despite varying hyperparameters, our method consistently delivers high performance, emphasizing its potential as a versatile solution in diverse real-world scenarios. However, care should be taken to appropriately tune these parameters according to the specifics of the given context.

Future work might explore other influential hyperparameters and conduct a similar sensitivity analysis. Moreover, utilizing methods like automatic hyperparameter tuning or evolutionary algorithms can lead to more efficient and optimal configuration settings.

Table 9: Performance of different hyperparameters.

| Metric | Maximum Iteration | | | | Auxiliary Explorer Sample Size | | | | Distribution Similarity Threshold | | | | Significance Threshold | | | |
|---|---|---|---|---|---|---|---|---|---|---|---|---|---|---|---|---|
| | 0.01 | 0.05 | 0.1 | 0.5 | 50 | 100 | 150 | 350 | 10 | 50 | 100 | 150 | 0.001 | 0.005 | 0.01 | 0.1 |
| Fidelity | 0.9996 | 0.9996 | 0.9996 | 0.9995 | 0.9996 | 0.9996 | 0.9996 | 0.9995 | 0.9996 | 0.9998 | 0.9998 | 0.9998 | 0.9996 | 0.9996 | 0.9996 | 0.9995 |
| Robustness | 1.0000 | 1.0000 | 1.0000 | 1.0000 | 1.0000 | 1.0000 | 1.0000 | 1.0000 | 1.0000 | 1.0000 | 1.0000 | 1.0000 | 1.0000 | 1.0000 | 1.0000 | 1.0000 |
| Correctness | 0.9992 | 0.9992 | 0.9992 | 0.9992 | 0.9993 | 0.9992 | 0.9992 | 0.9992 | 0.9992 | 0.9995 | 0.9996 | 0.9996 | 0.9992 | 0.9992 | 0.9992 | 0.9992 |
| Accuracy | 0.9992 | 0.9992 | 0.9992 | 0.9992 | 0.9993 | 0.9992 | 0.9992 | 0.9992 | 0.9992 | 0.9995 | 0.9996 | 0.9996 | 0.9992 | 0.9992 | 0.9992 | 0.9992 |
| TPR | 1.0000 | 1.0000 | 1.0000 | 1.0000 | 1.0000 | 1.0000 | 1.0000 | 1.0000 | 1.0000 | 1.0000 | 1.0000 | 1.0000 | 1.0000 | 1.0000 | 1.0000 | 1.0000 |
| TNR | 0.9844 | 0.9845 | 0.9849 | 0.9842 | 0.9853 | 0.9846 | 0.9845 | 0.9841 | 0.9844 | 0.9909 | 0.9925 | 0.9920 | 0.9850 | 0.9847 | 0.9848 | 0.9841 |

### A.5  Limitations

Though our method achieves global explanation with high fidelity, it still has several limitations. First, our method works well with tabular data but might be inapplicable to raw image data. It is because our method treats every dimension of the feature space as a semantic feature, such as the average packet size of a network connection, and extracts rules from each dimension of the feature space. In

contrast, raw image data are tensors of pixels. Their high-level semantics cannot directly derive from each of the pixels but usually need a deep model with spatial awareness (e.g., CNN, ViT) to extract feature maps, which are inconsistent with our method. Hence, this issue may limit the transferability of the proposed method to other domains. Nonetheless, due to the typical trust in expert knowledge over deep models in security domains, most security applications still rely on sophisticated feature engineering and use data representations with explicit semantics, suggesting that our method remains general in the field of security.

Second, recall that the rules extracted by our method are axis-aligned, which can be interpreted as a certain feature over/under a threshold and are human-understandable. Though this format of rules significantly promotes interpretability, it may limit its degree of fitting to the decision boundary of the original model, which can be of various shapes in the high-dimensional feature space for different models. Though our IC-Tree has mitigated this issue by splitting the distribution of normal data into multiple compositional distributions, which are more compact and more likely to be fitted using axis-aligned rules, there is little guarantee that our method cannot encounter underfitting if the decision boundary of the original model is extremely irregular. It should be clarified that this is a common limitation for all the global explanation methods that employ axis-aligned rules or decision trees as the surrogate expression. To this end, we are also exploring other surrogate models and algorithms that can further balance the interpretability and fitting ability.

Lastly, as we mentioned above, we believe that the experiments on "human understanding and trust" can be significantly strengthened by introducing security practitioners to participate in a use test, finding if the interpretation provided by the proposed method is consistent with their expert knowledge of judging an anomaly. Currently, we are collaborating with the Tencent Security Platform Department, aiming to rectify this limitation by accounting for the opinions of real practitioners.

