# OpenReview forum: "Interpreting Unsupervised Anomaly Detection in Security via Rule Extraction"
_NeurIPS.cc/2023/Conference — NeurIPS 2023 poster_

### Official Review · Reviewer_jw73 · 2023-06-15

**Soundness:** 2 fair
**Presentation:** 2 fair
**Contribution:** 3 good
**Rating:** 5
**Confidence:** 2

**Summary:**

This paper proposes a rule-set extraction method for a black-box anomaly detection model that is trained with only normal or unlabeled data.
There are many methods to interpret the learned model, but, this paper claims that there are few methods designed for anomaly detection.
The proposed method first extracts rules that decompose the complex distribution of normal data into multiple simple distributions and then, in each simple distribution, the decision rules that approximate the decision boundary of the pre-trained model are estimated.
By merging such rules, the proposed method can obtain a surrogate rule-based model of the trained anomaly detector.


**Strengths:**

- The problem tackled in this paper is important. There is a demand for interpretable anomaly detection in many industries, including security.
- The proposed method is model-agnostic and thus can be applied in many applications.
- The proposed method works well in the experiments.

**Weaknesses:**

- There is no discussion or comparison regarding computational costs.
- There are several unclear points about the methodology. This may be because I am a non-expert in this area (rule-based methods). I would like to wait for the author's additional explanation in the rebuttal.

**Questions:**

- This method is proposed as security-specific, but if I understand correctly, there is no security-specific part of the model. Is it better to present it as a more generic method since there are other areas of industry where rule sets are required?
- The proposed method is mostly feature-by-feature processing, so can it be used for high dimensional data as well? The datasets with small feature sizes (30) were used in the experiments. Some discussion of the computational cost would be required.
- Does Proposition 2 claim that the probabilities of the samples belonging to the same distribution $D_k$ are similar? If so, I think that the assumption is not satisfied, e.g., if $D$ is a Gaussian mixture distribution (i.e., $D_k$ is each Gaussian component). Please let me know if I have misunderstood.
- In Eq. (4), although $I(N_l)$ and $I(N_r)$ depend on $N_l$ and $N_r$, it is unclear how $N_l$ and $N_r$ are used for caluculating the value of $I$. I also did not understand why the output of anomaly detection could be used to partition the distribution in Eq. (4).
- In Eq. (8), It reads as the sum of a vector and scalar. Is the second term a vector?

Minors
- In section 1, this paper mentions one reason for using global explanatory methods is that forecasts are interpretable. However, I think that this property also be achieved by local explanation methods.
- It is better to explain the meaning of each loss of Eq. (2) in Section 2.
- In definition 3, $\phi_k$ is not defined.

**Limitations:**

Yes.

---

> ### Author Rebuttal · Authors · 2023-08-10
>
> We thank Reviewer jw73 for the detailed and constructive review. We will address all the expository comments in the final version of the paper.
>
> **Major clarification.**
>
> >Is it better to present it as a more generic method since there are other areas of industry where rule sets are required?
>
> Thanks for your comment, which we believe is constructive. Though we are not experts in other areas of industry, we do think our method can be applied to other areas that need rule-based decisions, as long as their data formats are tabular data where each column has explicit semantics. As we are more familiar with security applications that urgently need model interpretation, we use this as our starting point. Following your suggestions, we will add some examples of potential applications in other areas in the final paper and may conduct experiments in other areas in future work.
>
> >Can it be used for high-dimensional data as well?
>
> Yes. We have added a runtime experiment with different dimensions of data. Please refer to the global response.
>
> >Does Proposition 2 claim that the probabilities of the samples belonging to the same distribution Dk are similar?
>
> We are afraid that you may have misunderstood. This proposition actually states that if two data samples belong to the same compositional distribution $D_k$, the probability that they belong to the overall distribution of normal data ($D$) will not differ very much. Recall our definition of compositional distribution in Definition 3, the probability that a sample belonging to the compositional distribution $D_k$ belongs to other distributions is significantly small. As the probability of the overall distribution is equal to the weighted sum of the probability of each compositional distribution (also given in Definition 3), the sum of several values is close to 0 and $P_{X\sim D_k}(x)$ is still close to $P_{X\sim D_k}(x)$, which will not differ much for the two data samples belonging to $D_k$. This is indeed a bit similar to GMM as you mentioned, provided that different Gaussian components do not overlap with each other significantly.
>
> >It is unclear how $N_l$ and $N_r$ are used for calculating the value of $I$.
>
> Similar to CART decision trees, here $I$ is a criterion function to represent data impurity. We take the Gini index as an example: $I=2p(1-p)$, where $p$ originally is the probability of being normal for binary classification. In CART, this probability is estimated by the frequency of normal samples by counting their labels, i.e., $p_{N}=|\{ y\in N;y=0 \}| /|N|$. With the absence of labels in our scenario, we propose to use average output of anomaly detection to estimate this probability, as described in Eq. (5). For example, for $N_l$, the complete process of calculating $I(N_l)$ is: $ I(N_l)=2p_{N_l}(1-p_{N_l})$, where $p_{N_l}=\frac{1}{|N_l|}\sum_{x\in N_l}f(x)$.
>
> >I also did not understand why the output of anomaly detection could be used to partition the distribution in Eq. (4).
>
> The intuition behind our approach is that the original model is well-trained to understand the distribution of normal data well, and thus can output similar values for data from the same distribution. This intuition is a bit similar to knowledge distillation where the original model serves as a teacher. We let the output of the original model guide our partition of the data, and our goal is to find a partition of the space so that data points with similar output values from the original model are enclosed into one subspace.
>
> >In Eq. (8), It reads as the sum of a vector and scalar. Is the second term a vector?
>
> Yes, the second term should be a vector, or the first term should be the i-th dimension of the explorers. Since we used the i-th dimension as an example in the entire Section 5, when writing Eq. (8), we initially wanted to focus on the calculation of the i-th dimension, so Eq. (8) was written in the current form, which can be somewhat confusing. Thanks for your correction and we will modify it in the paper!
>
> **Minor clarification.**
>
> >I think that this property also be achieved by local explanation methods.
>
> Indeed, local explanation methods can certainly interpret one decision at a time. However, as mentioned in [a], local methods limit their attention to only a subset of individual decisions; they are prone to providing misleading explanations depending on the subset of samples analyzed, which may not be the best option for high-state applications like security.
> Besides, we also argue that global methods are better for achieving efficient online defense, as the extracted rules can be easily integrated with the majority of defense tools using rule-based expressions (e.g., iptables).
>
> [a] A. S. Jacobs, R. Beltiukov, W. Willinger, R. A. Ferreira, A. Gupta, and L. Z. Granville, “Ai/ml for network security: The emperor has no clothes,” in Proceedings of the 2022 ACM SIGSAC Conference on Computer and Communications Security. ACM, 2022.
>
> >It is better to explain the meaning of each loss of Eq. (2) in Section 2.
>
> Thanks for the suggestion. The two terms in Eq. (2) describe the fidelity of the extraction rule to the original model on normal data (i.e., $L_{X\sim D}(C,f,\varphi)$) and abnormal data (i.e., $L_{X\nsim D}(C,f,\varphi)$) respectively. Here $L(\cdot)$ can be any loss function suitable for binary scenarios, such as binary cross entropy. We will add more explanations in the paper for clarification.
>
> >In definition 3, $\phi_k$ is not defined.
>
> Thanks for your reminder. $\phi_k$ denotes the weight of each compositional distribution, and the overall distribution is obtained by the weighted sum of the compositional distributions.
>
> Again, thank you for your comments, and we hope that this addresses your concerns!

---

> > ### Comment · Reviewer_jw73 · 2023-08-16
> > **Thanks**
> >
> > I have read the rebuttal. Most of my concerns have been addressed.
> > In the revised version, I hope the authors discuss applications other than the security domain (even experimentation, if possible).
> > I think the problem addressed in this paper is important and the approach is reasonable.
> > Thus, I will increase my rating.

---

### Official Review · Reviewer_AbmJ · 2023-06-21

**Soundness:** 3 good
**Presentation:** 2 fair
**Contribution:** 3 good
**Rating:** 5
**Confidence:** 5

**Summary:**

The paper deals with the problem of "explainable" unsupervised machine learning (ML) for anomaly detection (AD), with a focus of network intrusion detection (NID). The paper argues that while abundant effort focused on providing _supervised_ ML techniques that are explainable, this is not the case for _unsupervised_ ML methods---which are abundant in NID due to the lack of fine-grained labelled data. The paper hence seeks to rectify such shortage by proposing a novel solution that simultaneously (i) provides "human-understandable" explanations; and (ii) allows to derive some "rules" that can be used to improve the detection performance of the unlderying network intrusion detection system (NIDS). The proposed method leverages the intuition of "slicing" the (training) data into segments that represent a given group of (network) activities, e.g., web or database traffic; and then derive specific rules that describe these activities in an "humanly understandable way" and which can be used (if violated) to identify anomalies. The method is rigorously described and its effectiveness is assessed through various experiments on two well-known NID datasets, showing some advantages over existing baselines (which are mostly tailored for supervised ML).

**Strengths:**

+ Evaluation on two NID datasets
+ Ablation Study
+ Rigorous description
+ Some design choices "make sense"
+ The method is rigorously described
+ Some ideas are correlated with proofs (in the appendix)
+ The paper tackles an open research problem (on the surface) for which limited research is available
+ Evaluation considering various parameter settings
+ The results show improvement over baselines


The paper addresses an intriguing research problem ("explainable unsupervised machine learning") which has high relevance in some real-world applications of ML (i.e., network intrusion detection). The proposed method is rigorously described: some intuitions are sensible, and for others there are proofs provided which increase the overall soundness of the proposed methodology. The evaluation (which is reproducible, since the code is disclosed and the datasets are publicly available) shows improvement over prior work (most of which, however, was not designed to work in the given setting).

Finally, the main novelty of this paper resides in addressing an underexplored problem.

**Weaknesses:**

## High Level

- Unconvincing motivation
- Excessively optimistic assumptions
- Flawed dataset and poor setup
- Questionable metrics
- Unclear "understanding" assessment

Despite some strengths, the paper suffers from various issues which prevent me from recommending inclusion in NeurIPS'23 program. Specifically, though the paper seeks to address a "real-world problem" (i.e., how security operators deal with ML-NIDS that are not explainable), the proposed method relies on assumptions that can hardly be met in the real world---thereby decreasing the overall significance of the findings.

Another problem pertains the soundness of the experimental procedure, which casts doubts on the validity of the results and, hence, of the conclusions drawn from the paper.

Finally, from a "novelty" perspective, the techniques used to address the problem tackled by this paper may be (somewhat) novel, but they can be redundant in light of some well-known considerations of the NID context.

## Low Level

I will provide an extensive description of the abovementioned weaknesses below. I will directly quote some statements made in the paper, and I will also attempt to provide suggestions and plenty of references to support my points.




### Unconvincing motivation

Albeit I acknowledge that the goal of providing "explainable ML" is relevant, I do not see a strong incentive in providing "explainable _unsupervised_ ML methods". This is because practitioners are well-aware that such methods are bound to be inaccurate, since they assume a notion of "normality". In a sense, the "explanation" for a raised anomaly is that a given data-point deviates from such normality; even practitioners acknowledge this as a fact [D]. This of course leads to a huge amount of false positives, which must be manually triaged _regardless of the explanations_.

In light of this, I did not find a compelling argument (provided in the paper) in favor of focusing on unsupervised ML --- or, at least, not in the context assumed by this paper. Perhaps a potential avenue is using said explanations for "attack attribution" [H], but this is never made explicit.

### Optimistic assumptions

The method makes some strong assumptions that are too optimistic in real settings. The problem is that the main contribution (allegedly) is addressed at real-world deployments of ML.

> We consider the anomaly detection tolerant of noisy data, but their proportion in training dataset is small and we do not have any ground truth labels of the training data.

This is a bold assumption which almost trivialized the entire problem of anomaly detection. If one is confident of having a (large) training set containing "certain" benign samples, one can use it as basis for supervised ML by enriching it with malicious samples (e.g., [I]). Unfortunately, in reality, obtaining such a "clean" training set requires huge manual effort [D] (and so is determining the threshold which allows for a manageable amount of alerts in operational contexts). Do note that the experimental evaluation assumes a "perfect" labelling of the normal samples in the training data (there is no mention of "noisy" data).

Also, consider the following statement

> For example, a server supports multiple services such as web, email and database. The representations of these services can be disparate and located in different regions in feature space with little transition between the regions, making it infeasible to find a uniform rule set to accurately estimate the original model

In theory, this is correct. In practice, however, addressing this problem is trivial: "web", "email" and "database" are all services that can be easily separated by looking at the source ports of the corresponding communications (potentially with an additional filter based on the IP address). Indeed, one must remember that these methods must be setup and deployed by network administrators---who are well aware of what services are running in their organization.

(note that even though here I have criticized the specific example, the consideration above can be extended to any type of potential "multimodality" that arises from network data)

In light of this, I fail to see any practical reason in favor of the method proposed in the paper. Perhaps the authors can provide a clear use-case in which the "multimodality" of the data is truly problematic to model.

### Flawed dataset (and poor experimental setup)

The proposed method is evaluated on two datasets for Network Intrusion Detection (NID), one being the CIC-IDS17. Unfortunately, this dataset is flawed [J, K]. Note that [J] came out in 2021, and it has already been well-received by the NID community, so it is concerning that this paper (which has been submitted to a top-venue such as NeurIPS) performs the experiments on the "flawed" variant of this dataset---especially given that a "fixed" version exists (provided in [J]).

Furthermore, note that [J] clearly stated that some features in this dataset (both the original and fixed version) are redundant for NID purposes (this is especially the case for "flow-based" NID [I, L]): however, this paper does not provide any detail about whether precautions have been taken to clean the feature set from "obvious" features (note that this also entails the ToN-IoT dataset)

Finally, it is unclear whether the experiments have been repeated a sufficient amount of times to derive statistically significant conclusions (as recommended in [B]). The paper states:

> The datasets are randomly split by the ratio of 6:2:2 for training, validation and testing

The "random split" may bias the results.

In light of this, the experimental setup may not be sound, thereby questioning the overall results. I invite the authors to look into other datasets as well (the works referenced in this review provide plenty of suggestions).


### Questionable evaluation metrics

I was unable to find a link between the metrics used in the evaluation and the overarching goal of the paper.

According to the Introduction:

> we can present the inferential process of the black-box model in a human-understandable way, and build a surrogate rule-based model for online defense at the same time.

However, there is nothing in the evaluation that allows one to measure whether the ```inferential process of the black-box model``` is presented in a ```human-understandable way```. This is because the evaluation appears to merely focus on the sheer detection performance of the "surrogate" model.

To re-quote a statement in the introduction (with which I fully agree!):

> security operators tend to trust human-understandable rules rather than the unintuitive outputs such as labels and numeric values from the complex and incomprehensible models.

There is little in the evaluation that makes me believe that security operators would appreciate the explanations provided by the proposed method.

I also have some concerns on the way these metrics are described:

> Fidelity (FD), i.e., the ratio of input samples on which the predictions of original model and surrogate model agree over the total samples, which indicates the extent to which humans can trust the explanations

I disagree: at best, FD denotes how much the surrogate and original models "agree". This says nothing about whether a human can "trust" the explanation in the general sense. Indeed, the quoted statement would be correct under the assumption that the original model's explanation are "correct".

> Robustness (RB), i.e, the persistence to withstand small perturbations of the input that do not change the prediction of the model

I am confused here: what is the ```model``` whose prediction should not change? I.e., is it the "original" or the "surrogate"?

### Unclear method for "understanding"

This issue follows the previous one. The "human-understandable way" to interpret the rules generated by the proposed method is assessed in Section 6.3. However, there is a lack of details in this Section, and the "conclusions" appear to be drawn from purely subjective statements.

Consider the following:

> Such results are easy to be interpreted.

How is this determined?

> These explanations are in line with how security experts recognize the attack data.

No reference is provided.

Plus, there is a mismatch with Table 4 and the overall assumptions made in the paper. Specifically, the paper assumes an "unsupervised" AD task, wherein the specific attack is not known. In light of this, how can a human derive that, e.g., (taken from Table 4):

> DDoS attacks use packets of small sizes to achieve
asymmetric resource consumption on the victim side,
and send packets at a high rate to flood the victim

? At most, a human sees some rules (e.g., ```ps_mean > 101.68, iat_mean > 0.063, dur > 12.61```) which -- when not met -- trigger some anomalies. But from here to stating that these anomalies relate to "DDoS attacks" there is a long way. For instance, "violation" of these rules can very well be related to a port-scanning activity (which can very well be a benign event, e.g., a new host which scans the network after joining it).


### Some bold statements in the Introduction

Below is a list of statements made in the Introduction for which I have serious concerns.


> it requires no attack/malicious data during the training (i.e., zero-positive learning), which are typically much more sparse and difficult to obtain in contrast with benign data;

Actually, obtaining "certain" benign data can be even harder than acquiring malicious data (see [B]). As a matter of fact, real-world deployments of ML entail very coarse "labelling" schemes [A].

> it does not fit any known threats, enabling better detection on unforeseen anomalies.

But this also means that some of the raised anomalies have nothing to do with attacks---leading to "alert fatigue" (see [C,D,E]).

> being self-explained and accurate for high-stake security applications.

Please define "accurate for high-stake security applications". Even non-surrogate models have a huge margin of error [C].

> Accuracy Loss (CH3). [...] In this case, though these methods can provide model explanation, they cannot meet the need of online deployment which requires high detection accuracy in security applications.

I disagree. The "surrogate" model has a single objective: provide "global" explanations. Therefore, even though such surrogate model may have worse "accuracy", it is the "larger" model that is meant to be deployed, while the "surrogate" is only used for explanation tasks.

> We observe that an important reason why simple surrogate models are ineffective is that they cannot learn well about the complex data distribution in high-dimensional space.

On what grounds is this "observation" made? Is it just an educated guess, does it derive from original experiments, or is it drawn from prior research?


> The extracted rules outperform prior work in terms of diverse metrics including fidelity, robustness, true positive rate and true negative rate

This contradicts what was written at the beginning of the introduction: ```security operators tend to trust human-understandable rules rather than the unintuitive outputs such as labels and numeric values from the complex and incomprehensible models.```. In what way are the metrics mentioned above different from "numeric values from the complex and incomprehensible models"?

> which meets the demand of improving human trust in black-box models and maintaining high detection accuracy for online deployment.

This is a self-claim: there is no evidence provided that ```fidelity, robustness, tpr, tnr``` meet the described ```demand``` (in terms of explainability).




### Some additional issues:

The following is a list of miscellaneous issues I found in this submission; note that these issues are not necessarily "minor".

* Some relevant works have not been considered. The extensive study carried out in [F] reveals, e.g., [G] -- which specifically focuses on providing "explainable unsupervised models".
* ```furthur``` typo
* I found the following statement to be misleading (and also unnecessary): ```we refer to adversarial attacks [44] and propose a method to approximate the optimal direction to explore the decision boundary```
* ```Further, most of their true positive rates cannot meet the requirement of using their rules for online defense, as the sheer amount of attack data they miss can cause huge damages.``` please provide facts instead of writing vague statements such as "can cause huge damages". Even a single false negative can lead to "huge damages" [D].
* I did not find any measure of the "runtime" used to train/test the proposed method.
* Did the evaluation _really_ take place on a server having an RTX3090 with 8GB of VRAM? (given that the RTX3090 has 24GB of VRAM?)
* Did the evaluation _really_ take place on a server having an Intel Xeon with only 8GB of RAM?
* To my understanding, the method assumes that the data-points represent "network flows" (such is the data contained in the considered datasets). However, there is a huge variability in the ways such flows can be generated (see [I,M]). What would happen if the netflow generator is different?

#### EXTERNAL REFERENCES

[A]: Van Ede, Thijs, et al. "Deepcase: Semi-supervised contextual analysis of security events." 2022 IEEE Symposium on Security and Privacy (SP). IEEE, 2022.

[B]: Apruzzese, Giovanni, Pavel Laskov, and Aliya Tastemirova. "SoK: The impact of unlabelled data in cyberthreat detection." 2022 IEEE 7th European Symposium on Security and Privacy (EuroS&P). IEEE, 2022.

[C]: Alahmadi, Bushra A., Louise Axon, and Ivan Martinovic. "99% False Positives: A Qualitative Study of {SOC} Analysts' Perspectives on Security Alarms." 31st USENIX Security Symposium (USENIX Security 22). 2022.

[D]: Apruzzese, Giovanni, et al. "The role of machine learning in cybersecurity." Digital Threats: Research and Practice 4.1 (2023): 1-38.

[E]: Hassan, Wajih Ul, et al. "Nodoze: Combatting threat alert fatigue with automated provenance triage." network and distributed systems security symposium. 2019.

[F]: Nadeem, Azqa, et al. "Sok: Explainable machine learning for computer security applications." arXiv preprint arXiv:2208.10605 (2022) [to appear in EuroS&P'23]

[G]: Wickramasinghe, Chathurika S., et al. "Explainable unsupervised machine learning for cyber-physical systems." IEEE Access 9 (2021): 131824-131843.

[H]: Nisioti, Antonia, et al. "From intrusion detection to attacker attribution: A comprehensive survey of unsupervised methods." IEEE Communications Surveys & Tutorials 20.4 (2018): 3369-3388.

[I]: Apruzzese, Giovanni, Luca Pajola, and Mauro Conti. "The cross-evaluation of machine learning-based network intrusion detection systems." IEEE Transactions on Network and Service Management (2022).

[J]: Liu, Lisa, et al. "Error Prevalence in NIDS datasets: A Case Study on CIC-IDS-2017 and CSE-CIC-IDS-2018." 2022 IEEE Conference on Communications and Network Security (CNS). IEEE, 2022.

[K]: Engelen, Gints, Vera Rimmer, and Wouter Joosen. "Troubleshooting an intrusion detection dataset: the CICIDS2017 case study." 2021 IEEE Security and Privacy Workshops (SPW). IEEE, 2021.

[L]: Arp, Daniel, et al. "Dos and don'ts of machine learning in computer security." 31st USENIX Security Symposium (USENIX Security 22). 2022.

[M]: Vormayr, Gernot, Joachim Fabini, and Tanja Zseby. "Why are my flows different? a tutorial on flow exporters." IEEE Communications Surveys & Tutorials 22.3 (2020): 2064-2103.

**Questions:**

I thank the authors for submitting their paper to NeurIPS'23. Despite my stark criticism, I liked reading it. This is why I endorse the authors in answering the following questions to the best of their abilities: if provided with compelling evidence, I may be willing to raise my score.

* Q1) Given that CIC-IDS17 is flaeed, how much time would it be necessary to re-do the experiments?
* Q2) What is the runtime required to train/test the proposed method, and how does it compare to existing methods?
* Q3) Are the details of the experimental platform correct?
* Q4) What features have been used to train the models?
* Q5) How was Table 4 derived?
* Q6) Have the experiments been carried out multiple times to accout for bias?
* Q7) Can the authors provide convincing evidence of the "necessity" of the proposed method _in practice_?
* Q8) How is the "robustness" measured?
* Q9) How long would it take to assess the efficacy of the proposed method under varying percentages of "noisy" data?
* Q10) Does the method _require_ netflow data to work? And, if so, how would the resulting performance change if the flows are generated in a different way?


Please refer to the main review for additional information

**Limitations:**

The "limitations" are discussed in the supplementary material. Unfortunately, they do not mention that, e.g., the proposed method relies on overly optimistic assumptions (for real deployments), nor that the "understanding" is likely drawn by subjective intuitions. Also, I see no reason for mentioning that the proposed method does not work well on images, unless this data-type becomes prevalent in NID (and I do not see this happening anytime soon).

==========

# Update after rebuttal

I have increased my score from a 3 (Reject) to a 5 (Borderline Accept); I have also increased the scores for Contribution and Soundness (from a 2 to a 3). The improvements are due to the paper now being much more truthful as to what it does, as well as by accounting for a "non-flawed" version of the considered datasets. Ultimately, I do not see any clear reason for "rejection", but I also do not feel strong reasons to accept this---aside from tackling a highly practical area of application of ML, which could serve as inspiration for future work.

---

> ### Author Rebuttal · Authors · 2023-08-10
>
> We thank Reviewer Abmj for the detailed and constructive review. We are very glad to hear so much constructive advice from a security expert like you. Many of the references are very helpful for NIDS research, which we would like to cite in our paper.
>
> >Given that CIC-IDS17 is flawed, how much time would it be necessary to re-do the experiments?
>
> We have tried our best to re-do the experiments on a new dataset as you suggest: CIC-IDS-improved. Please refer to the global response.
>
> >What is the runtime required to train/test the proposed method, and how does it compare to existing methods?
>
> We have added this experiment and please refer to the global response.
>
> >Are the details of the experimental platform correct?
>
> We are sorry for the mistakes in the experimental platform. Our server uses Intel(R) Xeon(R) Gold 5218 CPU @ 2.30GHz (128GB) and GeForce RTX 2080 Super (8GB). Note that GPU is only used for the training of some DL-based anomaly detectors (i.e., AE, VAE), and our rule extraction only requires the use of CPU. Thanks for your correction!
>
> >What features have been used to train the models?
>
> We use common flow-level features as shown in the following table, which consider both directions of 5-tuple flows and various flow statistics. Some of the original features are actually not used because 1) we find that they contain obvious errors, such as considerable numbers of NaN and infinity; 2) they are easy to create shortcut patterns due to specific testbed setup (i.e., spurious correlation mentioned in "Dos and don'ts of machine learning in computer security"), such as IP address. Considering that NeurIPS focuses on AI, we didn’t include too much details on network traffic in our paper. We will add more clarification in supplementary materials.
>
> |Attribute|Statistics|Direction|Number|
> |-|-|-|-|
> |packet count|-|forward, backward, bidirection|3|
> |packet size|mean/max/min/var|forward, backward, bidirection|12|
> |inter-arrival time|mean/max/min/var|forward, backward, bidirection|12|
> |flow duration|microsecond|bidirectional|1|
> |destination port|0-65535|forward|1|
> |L4 protocol|TCP,UDP,ICMP|-|1|
>
> >How was Table 4 derived?
>
> The violation of certain rules explains why the model makes predictions for anomalies. However, it can only serve as the necessary but not sufficient condition for an anomaly to be judged as a certain attack. Our workflow of integrating the explanation method with a NIDS is similar to [b]. When an anomaly is detected, the security operator still needs to inspect anomalies and judge the attack type based on his/her expert knowledge (i.e., “human understanding” in Table 4). At this stage, our rules can give some hints about how the model judges the anomalies by the violation of rules. However, the operator may find that the violated rules are far from the important features for an attack. It means there is an inconsistency between model decision logic and expert knowledge. In many cases, as mentioned in [b], it may be due to the underspecification of the model, such as shortcut learning. This indicates the model decisions cannot be trusted, and some efforts to upgrade the model are imperative (e.g., retraining, fine-tuning).
>
> >Have the experiments been carried out multiple times to account for bias?
>
> Yes, our experiments are actually done by 5-fold cross-validation. We first randomly split the dataset into 5 folds, and then pick up 3 folds for training, 1 fold for validation and 1 fold for testing (i.e., 6:2:2). With all possibilities of folds selected as training sets, each experiment is conducted by 10 times.
>
> >Can the authors provide convincing evidence of the "necessity" of the proposed method in practice?
>
> Yes. Due to the word limit, we will answer this question once the discussion phase starts.
>
> >How is the "robustness" measured?
>
> Following [a], the robustness is defined as the persistence of the *surrogate* model to withstand small perturbations of the input that do not change the prediction of the *original* model”. It is calculated as $\frac{\sum_{n=1}^N g(x_n)-g(x_n+\delta)}{N}$, where $g$ is the surrogate model (extracted rules) and $\delta$ denotes the small perturbation . In our code, since the features are normalized, $\delta$ is initially set to 0.01; if the addition of this perturbation changes the prediction of the original model, we will gradually decrease the value of $\delta$ until it satisfies the condition. It evaluates the fidelity of the rules to the original model when the original model is resistant to perturbations.
>
> >How long would it take to assess the efficacy of the proposed method under varying percentages of "noisy" data?
>
> Thanks for your suggestion. We are still making efforts on this experiment and aim to finish it within the next few days.
>
> >Does the method require netflow data to work? And, if so, how would the resulting performance change if the flows are generated in a different way?
>
> Yes, netflow data contained in the considered datasets are used. We acknowledge that various flow generators exist, which may employ different flow definitions and expiry mechanisms. However, our method itself is not tied to a particular flow generator. As a model interpretation method, the flow data applied by the extracted rules is always consistent with the original anomaly detection model. If the flows are generated in a different way from what the anomaly detection model expects, our rules are also likely to fail since they maintain fidelity to the original model. Thus, we think it may not be very fair to evaluate our rules using a different way of flow generation.
>
> Again, thank you for your comments. Due to the word limit, we will give more detailed feedback to answer other mentioned issues once the discussion phase starts!
>
> [a] Vilone, G. et al., A comparative analysis of rule-based, model-agnostic methods for explainable artificial intelligence, ICAC 2020
>
> [b] A. S. Jacobs et al., Ai/ml for network security: The emperor has no clothes, CCS 2022

---

> > ### Comment · Reviewer_AbmJ · 2023-08-11
> > **Ack**
> >
> > Dear authors,
> >
> > thank you for the detailed and professional response -- your effort is greatly appreciated.
> >
> > Given the length of your responses, I will go over each of them in detail during the next week (starting from Monday, August 14th).

---

> > ### Comment · Reviewer_AbmJ · 2023-08-15
> > **OK (but)**
> >
> > Dear authors,
> >
> > thank for your response. I've greatly appreciated the diligence in carrying out more experiments on the "fixed" version of CIC-IDS17. I also acknowledge the additional clarifications, as well as the fact that the experiments have been repeated many multiple times (although using "folds" may not tell the whole story). I'm looking forward to read the more detailed responses as well as seeing the new experiments on increased amounts of "noisy" data.
> >
> > My only lingering concern is w.r.t. Table 4. My question was aimed at the "human understanding" column of Table 4. To be precise: how was the text in Table 4 derived? Is it a formulation of the authors based on their own assessment -- or is it the result of some analysis stemming from NID experts/practitioners?
> >
> > =============
> >
> > On a side note, I found that this comment ```Considering that NeurIPS focuses on AI, we didn’t include too much details on network traffic in our paper.``` is a bit "convenient". It's true that NeurIPS focuses on AI, but **this paper** focuses on network traffic. It is therefore crucial to provide enough details (and, indeed, my criticism was mostly due to lack of such details), since the original paper trated this subject (which essentially sets this paper apart from most other NeurIPS submissions) in an extremely simplistic way.

---

> > > ### Author Response · Authors · 2023-08-20
> > > **Response to Your Concerns (Part 1)**
> > >
> > > As for your concern with respect to Table 4, we make the following clarifications.
> > >
> > > First, the “human understanding” column is indeed the assessment of our own researchers. As we mentioned in the rebuttal, we strongly acknowledge that real “user-study” with practitioners should be the most appropriate and intuitive way to evaluate “human trust/understanding”. Nevertheless, the absence of sufficient resources and connections with the industry indeed brings difficulties in conducting such experiments to us during this stage. And we also strongly endorse the reviewer’s opinion on the gap between “research” and “practice”. Specifically for the research on model explanation for security applications, we find that the evaluation of such “understanding” in many works (even on Top 4) does not explicitly describes how they obtain the “human understanding”, e.g., how many people are involved, their identities, how long they have been in the industry. For example, in [1], the authors present the explanation of two case studies using their framework called xNIDS: OS scan and HTTP flood, quoted from the paper:
> > >
> > > >Explanations: xNIDS identifies LLMNR, NBNS, and SSDP as the important features.
> > >
> > > >Network operators’ understanding: this result matches the well-known description of OS scan attacks, namely the attacker exploits various protocols to scan the network to look for possible vulnerabilities.
> > >
> > > However, they never mention who are the “network operators” providing such understanding. We assume that the logic here is to use the so-called “common” or “well-known” description of the attacks as human understanding, which can also be offered by the researchers themselves even if they do not have enough resources of practitioners. Similar issues also occur in another work [2]. Correspondingly, most of these research works put more effort into the comparison of numeric metrics (e.g., fidelity, stability).
> > >
> > > We are currently attempting to reach some practitioners. Though the number is still limited, we believe they will help us make the evaluation of our future work more solid.
> > >
> > > [1] F. Wei, H. Li, Z. Zhao, and H. Hu, “XNIDS: Explaining Deep Learning-based Network Intrusion Detection Systems for Active Intrusion Responses”, in 32nd USENIX Security Symposium, 2023.
> > >
> > > [2] D. Han et al., “DeepAID: Interpreting and Improving Deep Learning-based Anomaly Detection in Security Applications,” in Proceedings of the 2021 ACM SIGSAC Conference on Computer and Communications Security, 2021

---

> > > > ### Comment · Reviewer_AbmJ · 2023-08-20
> > > > **Quick ack**
> > > >
> > > > Dear authors,
> > > >
> > > > thank you for your response and transparency.
> > > >
> > > > In light of what you stated, I endorse you to **stress** that the sentences in Table 4 were derived from the interpretation of the authors, and hence may be subjective and not reflect the wide population of NIDS experts. However, a point should be made that doing this --despite its intrinsic limitations-- is "common" even in works accepted to top security venues---and that future works should aim to rectify this limitation by accounting for the opinion of experts.
> > > >
> > > > Would you be willing to acknowledge the above in the final paper?

---

> > > > > ### Author Response · Authors · 2023-08-21
> > > > > **Response**
> > > > >
> > > > > Thanks for your quick reply. Certainly, we strongly endorse that such claims are precise and necessary for a top venue like NeurIPS. We promise to acknowledge the claims and future work in the final paper to guarantee transparency. Thanks!

---

> > > ### Author Response · Authors · 2023-08-20
> > > **Response to Your Concerns (Part 2)**
> > >
> > > Besides, we are also going to add a step-by-step description of obtaining Table 4 for better clarity, which takes data points as input and then shows the steps that come up with the explanations. Here is an overview of the steps:
> > >
> > > 1. For a reported anomaly $x$, we use the IC-Tree to pinpoint the rule of normality that judges $x$ as anomalous. It is realized by inputting $x$ into the tree and recursively finding the leaf node. The rule is denoted by $C_x$.
> > >
> > > 2. For each constraint of features in $C_x$, we compare the corresponding feature value of $x$ with the range of the constraint.
> > >
> > > 3. For the feature values outside the range of the constraints, the Rules of Normality, Feature Values, and Feature Meaning (i.e., the three columns in Table 4) along with the raw data sample will be sent as the explanation to the security expert for further analysis.
> > >
> > > 4. The security expert will analyze the data sample to determine the type of attack (i.e., the Attack column in Table 4) and give her/his understanding of the important attributes that make her/him identify the attack (i.e., the Human Understanding column in Table 4).
> > >
> > > 5. If there is a huge gap between the provided explanation and expert understanding, it indicates that the anomaly detector is not trustworthy. The security expert may conduct further actions to improve the detector, such as retraining, fine-tuning, or reconsidering feature selection.
> > >
> > > Note that such a gap may not only occur when the anomaly detector makes erroneous decisions but also when the anomaly detector correctly detects an attack sample by unreasonable features. For example, the most obvious attribute of a Denial-of-Service (DoS) attack is typically its high rate of forwarding packets in order to overwhelm the victim. However, the provided explanation might suggest the IP address is an important feature for the anomaly detector to make this inference, which is quite doubtful from the perspective of a security expert. In practice, this may be due to “spurious correlation” or “shortcut learning” (mentioned in [3] and [4]). Specifically, for the aforementioned DoS example, it could be due to inappropriate testbed settings for the collection of training data, such as the DoS attack being launched from a separate host address while all other normal traffic is from other host addresses. We are willing to add more details in the final version of the paper.
> > >
> > > We again thank the reviewer for the valuable comments, and we believe, based on your comments, we can make our work more solid in the final version. If you still have any questions, we are always happy to reply.
> > >
> > > [3] A. S. Jacobs, et. al., “AI/ML for Network Security: The Emperor has no Clothes,” in Proceedings of the 2022 ACM SIGSAC Conference on Computer and Communications Security.
> > >
> > > [4] Arp, Daniel, et al. "Dos and don'ts of machine learning in computer security." 31st USENIX Security Symposium (USENIX Security 22). 2022.

---

> ### Author Response · Authors · 2023-08-10
> **More Responses to Your Concerns - Part 1**
>
> Hi Review AbmJ,
>
> Thanks again for your very detailed and constructive comments. Due to the word limit, we cannot finish answering some of your questions and mentioned issues, which we would like to answer during this discussion phase.
>
> ### About Motivation (related to Q7)
>
> The motivation of such an “explainable unsupervised ML method” for NID, as mentioned in our paper, mainly includes “trust over high stakes” and “online defense”. We would like to give a deeper explanation of these points to answer the reviewer’s concerns.
>
> **Trust over high stakes**.
> We acknowledge that anomaly-based NID (i.e., unsupervised) may not be as accurate as some SOTA supervised NID methods, as the reviewer mentioned. However, in recent years, the techniques of unsupervised NID have developed so much that they’ve been able to achieve remarkable detection accuracy on many attacks and sufficiently low false positive rates. For example, the latest unsupervised NIDS [1] conduct experiments on 48 attacks and can achieve an average AUC of 0.988 on brute force attacks, 0.974 on flooding attacks, 0.985 on web attacks, and 0.993 on malware traffic. Another new anomaly-based NIDS called HorusEye [2] can reduce the FPR to less than 0.05% while preserving a good TPR. Given these excellent efforts, unsupervised NID will be increasingly deployed in practical systems. For example, in [3], the authors have deployed their unsupervised NID after a WAF in a top Internet company. Though these new methods can achieve better accuracy, they can still make mistakes, and they still heavily rely on black-box models that we can’t intuitively understand. Therefore, proposing an explainable unsupervised ML method is imperative.
>
> **Online defense.**
> As we mentioned in the paper, rule-based models can be more easily integrated with the majority of defense tools than complex ML/DL models due to their high interpretability and high efficiency. For example, [4] achieves the automation of generating defense rules based on their interpretation results, including iptables and OpenFlow. Another point that we would like to express when mentioning “online defense”, is actually to directly use the extracted rule-based models as online NIDS. Though the reviewer mentioned that:
> > it is the "larger" model that is meant to be deployed, while the "surrogate" is only used for explanation tasks.
>
> In fact, it is possible to deploy the “surrogate”, as long as it preserves high fidelity to the original model. Especially with the emergence of programmable switches (P4 switch), rule-based models can be easily translated to flow rules for deployment and achieve extremely high throughput over GPU (e.g., Tbps). For example, [5] uses knowledge distillation to translate black-box models into P4 rules. But [5] can only realize the translation of supervised NID. HorusEye [2] first explores the deployment of unsupervised NID, but it is specific to iForest. Our proposed method can extract rules from any unsupervised NID, which can be very meaningful for realizing the general online deployment of complex unsupervised models in form of rules. This is also why we treat “fidelity” as an important metric for evaluation.

---

> > ### Comment · Reviewer_AbmJ · 2023-08-15
> > **Ack (Part 1)**
> >
> > Thank for the explanations. I acknowledge that recent years have seen an increased amount of "unsupervised" NIDS. However, this only applies to research: whether these methods will be deployed in practice (and will exhibit the same degree of effectiveness) is questionable (see the reference below).
> >
> > Henceforth, while I agree with the comment above, I invite the authors to clearly distinguish research from practice -- and, on this note, [3] is an excellent reference.
> >
> > ======
> >
> > Reference: [G. Apruzzese, P. Laskov and J. Schneider, "SoK: Pragmatic Assessment of Machine Learning for Network Intrusion Detection," in 2023 IEEE 8th European Symposium on Security and Privacy (EuroS&P), Delft, Netherlands, 2023 pp. 592-614.
> > doi: 10.1109/EuroSP57164.2023.00042]

---

> ### Author Response · Authors · 2023-08-10
> **More Responses to Your Concerns - Part 2**
>
> ### About Assumptions and Dataset
>
> In our assumptions, we make the following statement that the reviewer questioned:
>
> >We consider the anomaly detection tolerant of noisy data, but their proportion in the training dataset is small and we do not have any ground truth labels of the training data.
>
> First, we apologize for our expression which may confuse the reviewer. In fact, what we would like to express is that the training datasets **do contain noisy data**, which is consistent with the reviewer’s point of view. Based on this fact, we further assume that the proportion of noisy data is not significantly large, which is the basis of training unsupervised models. We believe this assumption is reasonable, as noisy data may come from outliers of extremely rare activities, imperfect flow processing, or attempting attacks, whose amount is trivial compared to normal traffic. In fact, these cases are also exactly some of the “flaws” in the original CIC-IDS datasets that wrongly label them as normal data (mentioned in [6]). It means that our experiments are never based on any assumptions of “perfect” normal samples (though we fully agree to redo the experiments using the improved datasets). Nevertheless, we still agree that experiments on different proportions of noisy data are necessary, as the reviewer suggested.
>
> Besides, this assumption is referenced from the definition of anomaly detection in [7], which the authors believe is "applicable to most real applications", quoted as follows:
>
> >"In this work, we consider the anomaly detection problem with noisy data, which means that ε > 0 though it is small, but we do not have the ground truth of which instances are abnormal or not. This definition is applicable to most real applications."
>
> Another statement that the reviewer didn’t quite agree with is our intuition of multimodality in network data. We appreciate the reviewer for the deep and professional reflection on this intuition. Indeed, our example might not be very convincing, as the reviewer pointed out that these services can be separated by ports, protocols, and addresses. However, there are still many cases where such multimodality exists while they are not easy to be separated by humans and need a method like ours to achieve data-driven clustering. For example, with the increasing prevalence of encrypted traffic, many useful header fields can be unavailable for network administrators, including addresses and ports (e.g., by VPN or Tor). This difficulty in identifying applications among encrypted traffic even prompts certain research, such as encrypted traffic classification [8] and website fingerprinting [9]. Another example of multimodality that might be more convincing, is the use of long-lived connections for different activities of IoT devices. In [10], the authors observe that an IP camera may use one single keep-alive flow with the remote cloud for both real-time video streaming and uploading clips when detecting movement, which can demonstrate highly different patterns (i.e., persistent vs burst) inside netflow data.

---

> > ### Comment · Reviewer_AbmJ · 2023-08-15
> > **Ack (Part 2)**
> >
> > I think it is a bit far-fetched to use the fact that CIC-IDS17 is "flawed" to suggest that the experiments were actually using "noisy data". Regardless, the impression was that this aspect was not properly addressed. The new experiments with varying amounts of noisy data is the best way to simulate such a (realistic) setting.
> >
> > With regards to the "multimodality", I appreciate the new examples, which make this aspect much more convincing (I was not aware of [10], thanks!)

---

> ### Author Response · Authors · 2023-08-10
> **More Responses to Your Concerns - Part 3**
>
> ### About “Understanding” and Metrics (related to Q5)
>
> Thanks for the comments. We would like to answer the reviewer’s concerns by explaining the characteristics of global explanation and the reason why we use the metrics in the paper for evaluation, and also providing some evidence of the metrics used by other works on global model explanation.
>
> **Global explanation.**
> Global explanations aim at describing how a given black-box model makes its decisions “as a whole”, quoted from [11]:
> >"such explanations typically take the form of an inherently interpretable model such as a rule set or a DT and become the main vehicles for studying the decision-making process of the original black-box model and examining its properties."
>
> Specifically, the interpretation of “decision-making process” here, or “inferential process of black-box model” in our paper, refers to the “rule set” or “a DT” themselves, as these rule-based surrogate models are inherently self-explained for their “inferential process”. For example, our method can output a rule set, where each rule defines the normal range of specific features; if some instance is identified as anomalous, the violation of certain rules can interpret the anomaly as “this feature of the instance does not resemble the pattern of normal data, thus it is judged as anomalous”, which is understandable for humans.
>
> **Ideal metric.**
> However, as the reviewer mentioned, there is still a gap between the “understandable inferential process” and “human’s trust in anomaly detection” during the evaluation. In fact, the most appropriate way to assess it should be employing a trial in which human security experts are invited to inspect the DTs/rules of interpretation, measuring the consistency of understanding. Yet this is not an easy task to conduct: to eliminate the subjective influence of each expert, enough people must be invited to participate in the trial, which is difficult for us to carry out.
>
> **Commonly used metrics.**
> To this end, the most commonly used metric becomes “fidelity”, which is not only used by our method and all five baseline methods but also used by most of the related works on model interpretation (e.g., [12-14]). We apologize that our description of the metric might be somewhat confusing. As the reviewer said:
> > This says nothing about whether a human can "trust" the explanation in the general sense. Indeed, the quoted statement would be correct under the assumption that the original model's explanation is "correct".
>
> However, the trust does not directly derive from making “correct” decisions; as a model interpretation method rather than directly building a classification model, the first goal is to **keep consistent** with the decisions of the black-box model using an interpretable surrogate model. Only if the interpretable rules have sufficient fidelity to the original model, a user can then inspect the rules to understand the decision logic of the original model, and determine whether to trust the model. For example, as mentioned in [11], a user may find a model susceptible to some issues (e.g., “shortcut learning” or “spurious correlations”), and then she/he decides whether to trust this model, update the model, or completely replace this model, which is all decided by users. In fact, we are also making attempts on proposing a “debugging” approach based on the interpretable rules, which guides users to fix the errors of the original model so that they can really “trust” the model.
>
> At last, we again greatly appreciate the reviewer for his/her extraordinary efforts in helping us improve the paper, and also feel glad that you enjoy reading our paper. We sincerely hope that our reply can answer your questions. If you still have any concerns, we are always happy to reply.

---

> > ### Comment · Reviewer_AbmJ · 2023-08-15
> > **Ack (Part 3)**
> >
> > Thank you for the response. This sentence essentially clarified what my main concern was:
> >
> > > Yet this is not an easy task to conduct: to eliminate the subjective influence of each expert, enough people must be invited to participate in the trial, which is difficult for us to carry out.
> >
> > I will be blunt: I have been following the "scene" of ML-based NIDS for a while, and despite dozens/hundreds of papers discussing this problem, the stark reality is that the state of practice is skeptical of most research -- including the one focused on "explainable" methods (as confirmed by the reference mentioned in my "Part 1 response", as well as by the one below this comment). I would have _extremely_ appreciated if the proposed method had been scrutinized via some "user-study" with practitioners. The lack of this study makes me dubious of the true effectiveness of the proposed method -- which essentially represents this paper main contribution.
> >
> > =====
> >
> > Reference: Alahmadi, Bushra A., Louise Axon, and Ivan Martinovic. "99% False Positives: A Qualitative Study of {SOC} Analysts' Perspectives on Security Alarms." 31st USENIX Security Symposium (USENIX Security 22). 2022.

---

> ### Author Response · Authors · 2023-08-10
> **More Responses to Your Concerns - References**
>
> [1] C. Fu, Q. Li, and K. Xu, “Detecting Unknown Encrypted Malicious Traffic in Real Time via Flow Interaction Graph Analysis,” in Proceedings 2023 Network and Distributed System Security Symposium
>
> [2] Y. Dong, et. al., “HorusEye: A Realtime IoT Malicious Traffic Detection Framework using Programmable Switches”, in 32nd USENIX Security Symposium USENIX Security 2023
>
> [3] R. Tang et al., “ZeroWall: Detecting Zero-Day Web Attacks through Encoder-Decoder Recurrent Neural Networks,” in IEEE INFOCOM 2020 - IEEE Conference on Computer Communications
>
> [4] F. Wei, H. Li, Z. Zhao, and H. Hu, “XNIDS: Explaining Deep Learning-based Network Intrusion Detection Systems for Active Intrusion Responses”, in 32nd USENIX Security Symposium USENIX Security 2023
>
> [5] G. Xie, et. al., “Mousika: Enable General In-Network Intelligence in Programmable Switches by Knowledge Distillation,” in IEEE INFOCOM 2022 - IEEE Conference on Computer Communications
>
> [6] Liu, Lisa, et al. "Error Prevalence in NIDS datasets: A Case Study on CIC-IDS-2017 and CSE-CIC-IDS-2018." 2022 IEEE Conference on Communications and Network Security (CNS). IEEE, 2022
>
> [7] M. Du, Z. Chen, C. Liu, R. Oak, and D. Song, “Lifelong Anomaly Detection Through Unlearning,” in Proceedings of the 2019 ACM SIGSAC Conference on Computer and Communications Security
>
> [8] C. Liu, L. He, G. Xiong, Z. Cao, and Z. Li, “FS-Net: A Flow Sequence Network For Encrypted Traffic Classification,” in IEEE INFOCOM 2019 - IEEE Conference on Computer Communications
>
> [9] J. Li et al., “Packet-Level Open-World App Fingerprinting on Wireless Traffic,” in Proceedings 2022 Network and Distributed System Security Symposium
>
> [10] R. Li, et. al., “IoTEnsemble: Detection of Botnet Attacks on Internet of Things,” in 27th European Symposium on Research in Computer Security Computer Security, ESORICS 2022
>
> [11] A. S. Jacobs, et. al., “AI/ML for Network Security: The Emperor has no Clothes,” in Proceedings of the 2022 ACM SIGSAC Conference on Computer and Communications Security.
>
> [12] Wu, Mike, et al. "Beyond sparsity: Tree regularization of deep models for interpretability." Proceedings of the AAAI conference on artificial intelligence. 2018.
>
> [13] Kazhdan, Dmitry, et al. "MEME: generating RNN model explanations via model extraction." arXiv preprint arXiv:2012.06954 (2020).
>
> [14] Li, Qiaomei, Rachel Cummings, and Yonatan Mintz. "Optimal Local Explainer Aggregation for Interpretable Prediction." Proceedings of the AAAI Conference on Artificial Intelligence. 2022.

---

> > ### Comment · Reviewer_AbmJ · 2023-08-15
> > **Conclusions**
> >
> > Overall, I am happy of the comments made by the authors, which clarified some of my concerns. I will definitely increase my score, but --currently-- I still do not believe that this paper represents a significant contribution to NeurIPS.
> >
> > However, I wonder whether the authors will provide the results of the experiments on varying amounts of noisy data. Can the authors do this before the end of the discussion phase?

---

> > > ### Author Response · Authors · 2023-08-17
> > > **Will Finish Experiments This Weekend**
> > >
> > > Dear Reviewer AbmJ,
> > >
> > > We are very glad that our efforts can clarify some of your concerns, and we really learn a lot from your valuable replies! For the experiments on varying amounts of noisy data, we are still running our code and expect to provide the results this weekend (before the end of the discussion phase, of course!). We refer to the settings of other papers and add 5 different proportions of noise data into the dataset. The experiments indeed take some time (i.e., 5 proportions * several datasets * training and calibration of 4 black-box models * multiple rounds of experiments). Some preliminary results seem that the proportions of noise data will not significantly impact the performance of our method. We promise to provide the complete results once we finish the experiments. Thanks!

---

> > > ### Author Response · Authors · 2023-08-18
> > > **Additional Experimental Results**
> > >
> > > We have just finished the additional experiments on varying amounts of noisy data, and we appreciate your patience. Referring to the experimental setting in [1], we evaluate two approaches for the injection of noisy data into the training set: 1) random noise; 2) data from other classes, i.e., attack data. Due to limited space, we primarily demonstrate the F1 score of the black-box models and the fidelity of our extracted rules to the black-box models; we have obtained all the metrics on the improved CIC-IDS datasets and will add them (e.g., robustness, tpr, tnr) in the appendix. The results are as follows (F1 score of black-box model/fidelity of rule extraction):
> > >
> > > |Proportion of Noise (random)|AE|VAE|OCSVM|iForest|
> > > |-|-|-|-|-|
> > > |0%|0.9740/0.9829|0.9733/0.9814|0.9645/0.9729|0.9745/0.9876|
> > > |1%|0.9761/0.9829|0.9763/0.9824|0.8621/0.9148|0.9782/0.9940|
> > > |3%|0.9821/0.9876|0.9698/0.9873|0.8287/0.8960|0.9907/0.9920|
> > > |5%|0.9735/0.9855|0.9714/0.9675|0.7896/0.9511|0.5356/0.7732|
> > > |10%|0.9748/0.9739|0.9704/0.9881|0.8128/0.9600|0.2162/0.5148|
> > >
> > > |Proportion of Noise (attack)|AE|VAE|OCSVM|iForest|
> > > |-|-|-|-|-|
> > > |0%|0.9671/0.9997|0.9644/0.9977|0.9623/0.9975|0.9628/0.9984|
> > > |1%|0.9732/0.9991|0.8756/0.9992|0.8260/0.9952|0.7386/0.9953|
> > > |3%|0.9476/0.9996|0.5232/0.9976|0.4885/0.9991|0.3207/0.9583|
> > > |5%|0.5392/0.9914|0.5566/0.9996|0.2926/0.9992|0.2994/0.9966|
> > > |10%|0.2126/0.9987|0.2566/0.9983|0.1047/0.9996|0.1254/0.9978|
> > >
> > > We find that with the increase of noisy data, the F1 scores of black-box models are gradually decreasing for most of the models. This accords with the basic intuition, as noisy data may perturb the distribution that an unsupervised model learns as normality, and the influence on the two ML models (OCSVM, iForest) is more obvious. For the fidelity of rule extraction, we find that the impact of the noisy data proportion is not significant: 36 of 40 fidelity scores in two tables preserve over 0.95, and the variation of fidelity scores is not obvious with the increase of noisy data for most of the models. This demonstrates that our rule extraction method can well achieve our design goal, which is always to obtain similar performance to the black-box model that it extracts from. Nonetheless, the results of iForest in the first table might reveal that a larger proportion of noisy data has a certain negative impact on the rule extraction for certain models.
> > >
> > > We again thank the reviewer for the replies, and we believe, based on your comments, we can make our work more solid in the final version. If you still have any questions, we are always happy to reply.
> > >
> > > [1] Sarah Itani, Fabian Lecron, Philippe Fortemps, A one-class classification decision tree based on kernel density estimation, Applied Soft Computing, Volume 91, 2020

---

### Official Review · Reviewer_eS8h · 2023-06-28

**Soundness:** 3 good
**Presentation:** 3 good
**Contribution:** 3 good
**Rating:** 6
**Confidence:** 3

**Summary:**

The authors propose and Distribution Decomposition Rules and Boundary
Inference Rules to make black boxes more interpretable.

They use Interior Clustering Tree IC-tree to find distribution
decomposition rules.  The IC-tree algorithm splits the data on a
feature value at each node recursively.  The feature value that
maximizes the gain in gini index before and after the split is chosen
at each node.  Gini index is 2p(1-), where p be is the average output
of the anomaly detection model f(x), which is the trained neural
network model.  The path to each leaf forms a distribution
decomposition rule and each leaf represents a data subset that is in
the same distribution.

The Compositional Boundary Exploration (CBE) algorithm finds a minimal
hypercube that encloses the normal data in each distribution (leaf
node).  CBE uniformly samples data points on each hyperplane as initial
explores (centriods), then samples from a Gaussian distributions
points as auxiliary explorers near the centroids within some radius.
The samples are used to query the model to find samples with the
lowest probability to be normal (negative examples), these are called
candidate explorers.  They use Fast Gradient Sign Method to perturb
the explorers.  Since the loss function is not known from the model,
they approximate the gradient by the gradient between the model
outputs of an initial explorer and an auxiliary explorer with respect
to the different in the samples.  The explores are perturbed until the
f() indicates it is normal (less than threshold phi).  The
corresponding feature values forms the hyperplanes for the boundaries.

They evaluate 5 existing extraction methods with 4 black-box models and
2 datasets.  In addition to TPR and TNR, they measure Fidelity (which
calculates agreement between the black-box model and their extracted
rules) and Robustness (which estimates small input perturbations do not
change the model's output).  Their proposed method generally
outperforms existing techniques.  Ablation studies were performed to
access the contribution of each component.

I have read the authors' responses and commented on them.

**Strengths:**

1.  The proposed algorthms yield interpretables rules with high
    fidelity of the black-box models.

2.  Empirical results indicate the proposed algorithms generally
    outperform existing ones.

3.  The paper is generally well written.

**Weaknesses:**

1.  Some ideas could be clarified, see questions below.

**Questions:**

1.  line 220: What is the reasoning for the initial vs auxiliary explorers?

2.  line 227: What is the reasoning for "minimal probability of being normal"?
     Why not choose candidates that are close to the threshold $\varphi$ and then
     try to get to the other side of the threshold?

3.  line 227: Does each of the dimension/feature have two candidate
    explorers so that they yield two boundary (axis-parallel
    hyperplane) in each dimension?  How do you guarantee one boundary
    on each side in one dimension?

4.  line 239: should f() be greater than $\varphi$ instead of less than?  I think
     you try to go from low f() to high f() to cross $\varphi$.

5.  line 323: "directly using hypercubes as rules", how are these
    hypercubes found?  Are they from the IC-tree?  A hypercube can
    be found for instances in each leaf by finding the "min and max" value
    of each feature/dimension via a fitted Gaussian.  If that is not what you did, adding
    it as a comparison would be interesting.

6.  line 325: "Though using the K-Means can reach similar results, it
    cannot be expressed by axis-aligned rules with high interpretability
    and deployability as the IC-Tree can achieve."

    Using CBE, axis-aligned rules can be found with clusters found by
    K-means.  If I understand correctly, the IC-tree can be ignored
    after the boundary rules are extracted by CBE.

**Limitations:**

Limitations of the proposed approach seem to be discussed as
directions in future work.

---

> ### Author Rebuttal · Authors · 2023-08-10
>
> We thank Reviewer eS8h for the detailed and constructive review, especially thanks for your recognition of our work. We will address all the expository comments in the final version of the paper.
>
> >line 220: What is the reasoning for the initial vs auxiliary explorers?
>
> The initial explorers set the starting point for the search on each hyperplane. Since the range of some hyperplanes may be large, and we do not know from which point of the hyperplane we can find the decision boundary of the model the fastest, we evenly sample $N_e$ initial explorers on the hyperplane to increase the possibility that we can successfully explore the decision boundary.
>
> The auxiliary explorers are further sampled near each initial explorer. Their role is to explore the decision boundary of the model using an initial explorer as a starting point. We will eventually find the $N_e$ points closest to the decision boundary from all the auxiliary explorers, through the output of the black-box model and the Beam Search method, and generate the next iteration of the initial explorers based on them.
>
> >line 227: What is the reasoning for "minimal probability of being normal"? Why not choose candidates that are close to the threshold $\phi$ and then try to get to the other side of the threshold?
>
> Thanks for your comment. In fact, your understanding is consistent with our approach, and probably it’s our expression that confuses you. In Definition 1, we abstract an unsupervised anomaly detection model to a probability density function of being normal. Therefore, a minimal model output that is close to the threshold is equivalent to a "minimal probability of being normal". We apologize for our confusing expressions and will give more clarification in the paper.
>
> >line 227: Does each of the dimension/feature have two candidate explorers so that they yield two boundary (axis-parallel hyperplane) in each dimension? How do you guarantee one boundary on each side in one dimension?
>
> Yes, the boundary exploration will be conducted on each side of one dimension so that two boundaries (one upper bound and one lower bound) are expected to be obtained; your understanding is exactly correct. For your second question, though we expect to obtain one boundary on each side in one dimension, in fact, we do not guarantee the acquisition. As we stated in line 246, some of the dimensions might be a contour line for the output of the anomaly detection model, i.e., there is no correlation between model decisions and this specific feature. In this case, we just leave this dimension “open” and do not generate any constraint on this dimension.
>
> >line 239: should f() be greater than $\varphi$ instead of less than? I think you try to go from low f() to high f() to cross $\varphi$.
>
> Thanks for your comment. Again, we think it’s our expression that confuses you. Due to our abstraction of an unsupervised anomaly detection model to a probability density function of being *normal*, the lower model output suggests more anomalous under this setting. This may indeed go against our common sense a bit, such as MSE as a model output when a higher value represents a more anomaly.
>
> >line 323: "directly using hypercubes as rules", how are these hypercubes found? Are they from the IC-tree? A hypercube can be found for instances in each leaf by finding the "min and max" value of each feature/dimension via a fitted Gaussian. If that is not what you did, adding it as a comparison would be interesting.
>
> Yes, your understanding is exactly correct. A hypercube is found on each of the leaf nodes of the IC-Tree via the classic min/max approach as you mention.
>
> >Using CBE, axis-aligned rules can be found with clusters found by K-means. If I understand correctly, the IC-tree can be ignored after the boundary rules are extracted by CBE
>
> Thanks for your valuable question. Indeed, as you point out, after obtaining the final rule set, the IC-Tree can be ignored. However, an issue here is that, as defined in Eq. (3), the final rules are the conjunction of distribution decomposition rules (obtained by IC-Tree) and boundary inference rules (obtained by CBE) for each compositional distribution. If we use clustering algorithms like K-Means to substitute IC-Tree, we cannot obtain axis-aligned rules anymore that can be combined with boundary inference rules obtained by CBE, which hinders the implementation of end-to-end axis-aligned rule extraction for better interpretability.
>
> Again, thank you for your comments, and we hope that this addresses your concerns!

---

> > ### Comment · Reviewer_eS8h · 2023-08-17
> > **comments on response**
> >
> > Thanks for your response.
> >
> > >>    Using CBE, axis-aligned rules can be found with clusters found by K-means. If I understand correctly, the IC-tree can be ignored after >>the boundary rules are extracted by CBE
> >
> > >Thanks for your valuable question. Indeed, as you point out, after obtaining the final rule set, the IC-Tree can be ignored. However, an issue > here is that, as defined in Eq. (3), the final rules are the conjunction of distribution decomposition rules (obtained by IC-Tree) and boundary > inference rules (obtained by CBE) for each compositional distribution. If we use clustering algorithms like K-Means to substitute IC-Tree, we > cannot obtain axis-aligned rules anymore that can be combined with boundary inference rules obtained by CBE, which hinders the
> > > implementation of end-to-end axis-aligned rule extraction for better interpretability.
> >
> > After clustering, for each cluster, one can find a minimum hypercube by finding the min and max in each dimension that encloses the instances in the cluster.  Using these min and max values in each dimension, one can form a rule for the hypercube.  What are the disadvantages?

---

> > > ### Author Response · Authors · 2023-08-18
> > > **Response to Your Questions**
> > >
> > > We thank the reviewer for the in-depth reflection on various possibilities for building a rule extraction algorithm. The way that the reviewer describes to translate clusters into hypercubes by min/max values to serve the distribution decomposition rules is indeed practical. Note that the distribution decomposition rules actually constrain the subspaces explored by the CBE algorithm, as suggested by Eq. (3) (i.e., the final rule is the conjunction of a distribution decomposition rule and a boundary inference rule). Directly using the hypercube derived from the cluster, however, will drastically reduce the subspace as the hypercube is compact, which then hinders the area that the CBE algorithm can explore. It brings the disadvantage that the final extracted rules will be more overfit to the normal data appearing in the training set, which gives the rules good detection capability of anomalies, while may significantly decrease the generalization of identifying other normal samples, resulting in a higher false positive rate. As mentioned in [1], high false alarm rates will cause “alert fatigue” and will significantly sabotage the usability of security applications.
> > >
> > > To verify the anticipation, we conduct an additional experiment using the mechanism suggested by the reviewer. The clustering algorithm is K-Means where k is set to 5 and 10. The results of fidelity on positive data and negative data are as follows:
> > >
> > > |Dataset|Method|Fidelity (Positive)|Fidelity (Negative)|
> > > |-|-|-|-|
> > > |CIC-IDS|K-Means (to rule) + CBE (k=5)|1.0|0.5299|
> > > |CIC-IDS|K-Means (to rule) + CBE (k=10)|1.0|0.0|
> > > |CIC-IDS|IC-Tree + CBE (ours)|0.9994|0.9967|
> > > |TON-IoT|K-Means (to rule) + CBE (k=5)|1.0|0.6601|
> > > |TON-IoT|K-Means (to rule) + CBE (k=10)|1.0|0.0|
> > > |TON-IoT|IC-Tree + CBE (ours)|0.9819|0.9941|
> > >
> > > Basically, the results are consistent with our anticipation. Again, we thank the reviewer for the valuable question. We think that adding this experiment as part of our ablation study might be a good idea. If you still have any concerns, we are always happy to reply.
> > >
> > > [1] Hassan, Wajih Ul, et al. "Nodoze: Combatting threat alert fatigue with automated provenance triage." network and distributed systems security symposium. 2019.

---

> > > > ### Comment · Reviewer_eS8h · 2023-08-19
> > > > **comments on response**
> > > >
> > > > Is the # of clusters (k=5 and 10) similar to the number of leaves in the IC-Tree? That is, the number of rules are consistent between k-means and IC-Tree?

---

> > > > > ### Author Response · Authors · 2023-08-20
> > > > > **Response to Your Question**
> > > > >
> > > > > Thanks for your question. Yes, actually we have conducted experiments on various settings of clusters for fair comparison between the number of clusters and leaves. For example, the number of leaves in the IC-Tree for the TON-IoT dataset is 17. Accordingly, we verified the value of k from 5 to 25 for the experiment of “K-Means (to rule)”. As the following results show, after k=10, the fidelity on negative data stays at 0. This is consistent with our anticipation: using more clusters will lead to more compact subspaces for the CBE algorithm to explore, which is more likely to overfit the training data points while making false positives during the testing.
> > > > >
> > > > > |#Cluster (k)|Fidelity (Positive)|Fidelity (Negative)|
> > > > > |-|-|-|
> > > > > |5|1.0|0.6601|
> > > > > |10|1.0|0.0|
> > > > > |15|1.0|0.0|
> > > > > |20|1.0|0.0|
> > > > > |25|1.0|0.0|

---

### Official Review · Reviewer_mhUf · 2023-07-05

**Soundness:** 3 good
**Presentation:** 3 good
**Contribution:** 3 good
**Rating:** 5
**Confidence:** 4

**Summary:**

The paper presents a technique to build a decision tree (DT) that uses the predictions from any unsupervised anomaly detection algorithm to split at the nodes when constructing the DT. This is named as the Interior Clustering Tree. The DT will be used to extract interpretable rules.

**Strengths:**

1. The approach is useful for explainable anomaly detection and is an important problem domain.

**Weaknesses:**

1. Very similar to [1]. The difference is that in [1] the density is based on KDE, whereas, in the present work the density is based on the anomaly detector. So the technical novelty is less.

2. Section 5: Lack of clarity; the procedure should be presented as algorithmic steps.

3. We need more than just two datasets for empirical validity. These datasets also seem 'easy' since detection accuracy is very high for all anomaly detection algorithms. With easy datasets, the rules extracted would tend to be short and nicely interpretable. On the other hand, with harder datasets, the rules might be much more complex and difficult to interpret. I am interested in the types of rules extracted for the harder datasets.

References:

[1] Sarah Itani, Fabian Lecron, Philippe Fortemps, A one-class classification decision tree based on kernel density estimation, Applied Soft Computing, Volume 91, 2020 (https://arxiv.org/abs/1805.05021)

**Questions:**

1. What is the order of computational complexity or the execution time of the algorithm? Is the also affected by the curse of dimensionality?

---

> ### Author Rebuttal · Authors · 2023-08-10
>
> We thank Reviewer mhUf for the detailed and constructive review. We will address all the expository comments in the final version of the paper.
>
> > Very similar to [1].
>
> Thanks for your comment. We have thoroughly read the provided paper, which proposes the one-class decision tree (OC-Tree). Though it may have some similar design points to ours, we did not refer to this work at all, and there is great difference between the provided paper and our work in terms of both purposes and methods.
>
> **Different purposes.**
> The OC-Tree is essentially a model for one-class classification (OCC), which is designed to find the distribution of the one-class data by enclosing the data within leaf nodes. Our work is different because our purpose is to extract interpretable rules from black-box anomaly detection models using one-class data. As an interpretation method, our focus is not on directly understanding the distribution of one-class data itself, but on explaining how the black-box model cognizes the distribution within the feature space, that is, how to replace the nonlinear decision boundary of the black-box model with interpretable rules, as suggested in Eq. (2) in our paper. This is also why an important evaluation metric for us is the fidelity of the black-box model, rather than just focusing on the accuracy of extracted rules for OCC.
>
> **Different methods.**
> Our method basically consists of the Interior Clustering Tree (IC-Tree) and the Compositional Boundary Exploration (CBE) algorithm. For IC-Tree, though it may have some similarities with OC-Tree in design, such as tree-based structure and new splitting mechanisms, it is significantly different from OC-Tree in that, as the reviewer indicates, OC-Tree relies on density estimation, while our IC-Tree uses the output of the anomaly detector as an indicator to partition feature space. A deeper understanding of this difference is that our IC-Tree does not come directly from the distribution of data in Euclidean space, but is guided by the black-box anomaly detector and tries to find the correlation between the decision-making logic of the black-box model and Euclidean space. Hence, our IC-Tree is different from OC-Tree from the starting point of the design.
>
> Second, our CBE algorithm is a completely absent part of OC-Tree. This is also essentially due to the difference in purpose: OC-Tree only needs to find data distributions with a high density, while our method needs to continue to estimate the decision boundary of the black-box model on the basis of concentrated data distribution.
>
> In summary, we think that our work and OC-Tree are still very different, and we believe that our approach still has innovative contributions to the community.
>
> > Section 5: Lack of clarity; the procedure should be presented as algorithmic steps.
>
> Thanks for your valuable suggestion. Due to limited space, we didn’t include an overall algorithmic process for the CBE algorithm in the manuscript. We will add it to the paper. At this stage, we give a preliminary version of the algorithm using pseudocode as follows:
>
> ---
>
> Input: data falling into k-th leaf node $X_k$, anomaly detector $f$ and its threshold $\varphi$
>
> Output: Boundary Inference Rule $C_k$ on this leaf node
>
> $H_k \gets$ MinimalHypercube($X_k$);
>
> for i-th dimension in $X_k$:
>
> $\ \ $while True:
>
> $\ \ \ \ $$e^{(1)},..., e^{(N_e)}\gets$ InitialExplorer($H_k$);
>
> $\ \ \ \ $for each initial explorer $e$:
>
> $\ \ \ \ \ \ $$\hat{e}^{(1)},...,\hat{e}^{(N_s)}\gets$ AuxiliaryExplorer($e$);
>
> $\ \ \ \ $Beam Search for $N_e$ number of auxiliary explorers with minimal probability of being normal judged by $f$ and $\varphi$;
>
> $\ \ \ \ $$e\gets$GradientApprox($\hat{e}$) for each $\hat{e}$ of the $N_e$ auxiliary explorers;
>
> $\ \ \ \ $if it satisfies ending condition:
>
> $\ \ \ \ \ \ $$c_i\gets (x_i \odot \hat{e}_i)$; break;
>
> return $C_k\gets H_k\vee(c_1\wedge c_2\wedge ...)$;
>
> ---
>
> > We need more than just two datasets for empirical validity. These datasets also seem 'easy' since detection accuracy is very high for all anomaly detection algorithms.
>
> Thanks for your valuable comment. We would like to resolve your concerns with both clarification and additional experiments.
>
> **Clarification.**
> First, although we stated in the paper that only two datasets are used for evaluation, each of them actually consists of multiple datasets in which different types of network attacks are collected. For example, TON-IoT datasets contain 9 types of network attacks, including backdoor, DoS, DDoS, injection, man-in-the-middle, brute force login, ransomware, scanning, and cross-site scripting. To ease the demonstration, the results in Table 3 in the paper are the average values of different attack datasets. We have added the result of each attack dataset in the attachment of the global response.
>
> Second, these datasets seem simple because security applications often rely heavily on feature engineering based on expert knowledge, so the features contained in the datasets have a high degree of discrimination. Other works published at top conferences, such as references [1-3] in the paper, also achieve very high anomaly detection accuracy (over 0.99). The characteristics of datasets may be somewhat different from those of other domains. For example, datasets in the CV domain typically only contain raw pixels.
>
> **Additional experiments.**
>
> We have tried our best to add the evaluation on a new dataset that contains more features. Please refer to the global response. We are still conducting experiments on another dataset and will add the results in the paper.
>
> >What is the order of computational complexity or the execution time of the algorithm? Is the also affected by the curse of dimensionality?
>
> We have added the evaluation to the global response.
>
> Again, thank you for your comments, and we hope that this has addressed your concerns!

---

> > ### Comment · Reviewer_mhUf · 2023-08-17
> > **Security Dataset**
> >
> > I thank the authors for the response to my comments.
> >
> > 1. Given the generic nature of the algorithm, there is really no good reason to restrict the experiments to security domain as it artificially constraints the number of datasets. Even so, for another security domain dataset you might consider the DARPA Transparent Computing dataset (https://github.com/darpa-i2o/Transparent-Computing). This has been cited, for example, in [1].
> >
> > 2. From the rebuttal: "it is significantly different from OC-Tree in that, as the reviewer indicates, OC-Tree relies on density estimation, while our IC-Tree uses the output of the anomaly detector as an indicator to partition feature space" -- I do not agree. The density estimate often can be used as a proxy for the anomaly score and could be considered as the output of the anomaly detector.
> >
> > 3. While I appreciate that the authors have presented an algorithm in the rebuttal, it could still be improved upon. The main issue I have here is that it becomes hard to tell whether the boundary we are trying to compute is trying to exclude anomalies while encapsulating the normal points, or is it the other way around. I know it is the former, but it needs to be reinforced a bit more. E.g., the output of the algorithm should defined as: "Output: Boundary Inference Rule C_k on this leaf node such that C_k encapsulates normal data".
> >
> > Also it would help readability to have a simple description for the algorithm (maybe in the caption).
> >
> > In addition, I agree with another reviewer's comment that the 'Understanding Model Decisions' section lacks detail. For clarity, it would be good to illustrate an algorithm that takes as input any new data point and then shows the steps that come up with the Table 4 explanations.
> >
> > References:
> > [1] Siddiqui, Md Amran, et al. "Feedback-guided anomaly discovery via online optimization." Proceedings of the 24th ACM SIGKDD international conference on knowledge discovery & data mining. 2018.

---

> > > ### Author Response · Authors · 2023-08-18
> > > **Response to Your Questions (Part 1)**
> > >
> > > We thank Reviewer mhUf for the reply to our rebuttal.
> > >
> > > ### About Datasets
> > >
> > > Indeed, as you and Reviewer jw73 suggested, our algorithm is inherently generic. We thank you for this very constructive suggestion. At the current stage, as our entire method is fully motivated by the requirements of security applications, our design goals and evaluation are therefore more specific to security domains. For example, to meet the requirement of “online defense” (mentioned in Section 1) that directly relies on the efficacy of the surrogate model (i.e., extracted rules), we need to guarantee that the extracted rules will not generate noticeably more false alarms than the black-box models (formulated by the first item in Eq. (2)), which is very important to the usability of security applications (i.e., “alert fatigue” in [1]) while may or may not be that significant to applications in other domains. To achieve this goal, our CBE algorithm is designed to *explore* the utmost boundary of normal data on a leaf node rather than directly using a hypercube to encompass the normal data, which can reasonably enlarge the area of normality to reduce false alarms. This is also why we include TNR (i.e., 1-FPR) as a metric for evaluation, while some other works on model interpretation do not evaluate the TNR (e.g., one of our baselines [2]). Nevertheless, we are very willing to add more discussions about the application to other domains in the final version, e.g., by experimentation using the datasets of other areas, to demonstrate the transferability of our method.
> > >
> > > We also greatly appreciate the reviewer for recommending the new dataset. However, after thoroughly investigate the dataset, we find conducting experiments on this dataset is a bit tricky, due to several stringent system requirements, e.g., using certain Bash scripts to load data from compressed files, loading annotations. Besides, this dataset is very large (100+ files * 250MB approximately * 6 subsets $\approx$ 150 GB in total). Therefore, it may not be practical to complete evaluation on this dataset before the end of this rebuttal phase. Further, by reading the description provided by the dataset, we find that one of their data formats is *netflow*. This is one of the most commonly used trace collection approaches for networking datasets (discussed in [3]), which is exactly consistent with the data format of our datasets used in the paper. Therefore, we believe that our evaluation has demonstrated the effectiveness of our method on such types of datasets. Nevertheless, we are more than willing to explore this valuable dataset in our future work.

---

> > > ### Author Response · Authors · 2023-08-18
> > > **Response to Your Questions (Part 2)**
> > >
> > > Besides the dataset that we evaluated during our first response in the rebuttal phase, we just finished the evaluation of another new dataset, as we promised in the rebuttal. This dataset is recommended by another reviewer (improved CSE-CIC-IDS2018 in [4], 36.04GB, 15 labels, 80 features). We conducted similar experiments to the ones in our paper, i.e., using 6 different methods (including baselines) to extract rules from 4 black-box unsupervised models trained on this dataset, and evaluating 4 metrics of the extracted rules. The results are as follows:
> > >
> > > | Method  |   AE       |          |         |         |   VAE      |          |         |         |   OCSVM   |          |         |         |   iForest |          |         |         |
> > > | ------- | ------- | -------- | ------- | ------- | ------- | -------- | ------- | ------- | ------- | -------- | ------- | ------- | ------- | -------- | ------- | ------- |
> > > |         |   FD      | RB       | TPR     | TNR     |   FD      | RB       | TPR     | TNR     |   FD      | RB       | TPR     | TNR     |   FD      | RB       | TPR     | TNR     |
> > > | UAD     |   0.0045  | 0.5      | 0       | 1       |   0.0045  | 0.5      | 0       | 1       |   0.9847  | 0.4885   | 0.9851  | 0.8865  |   0.9664  | 0.4655   | 0.9268  | 0.8885  |
> > > | EGDT    |   0.0063  | 1        | 0.002   | 0.9714  |   0.0044  | 0.9999   | 0.0002  | 0.9468  |   0.9955  | 1        | 1       | 0       |   0.7006  | 0.8374   | 0.7012  | 0.5767  |
> > > | Trustee |   0.0062  | 0.9981   | 0.0018  | 0.9796 |   0.0046  | 0.9996   | 0.0002  | 0.9703  |   0.0626  | 0.0669   | 1       | 1       |   0       | 0.9956   | 0.9479  | 0.1042  |
> > > | LIME    | 0.5708 | 0.9152 | 0.8126 | 0.0316 | 0.7356 | 0.9084 | 0.9215 | 0.0832 | 0.0517 | 1.00 | 0.9876 | 0.0484 | 0.8407  | 1      | 0.9651 | 0.9417 |
> > > | KD      |   0.0045  | 0.9960        | 0       | 1       |   0.0045  | 1        | 0       | 0.9928  |   0.0545  | 0.9999   | 0.053   | 0.3753  |   0.0045  | 0.0018   | 0       | 1       |
> > > | Ours    |   **0.936**   | **0.9997**   | **0.9359**  | 0.9611  |   **0.9357**  | 0.9991   | **0.9656**  | 0.9719  |   0.9356  | **1.00**   | **0.9855**  | **0.9811**  |   **0.9703**  | 0.9997   | **0.97**    | 0.9747  |
> > >
> > > The results show that our method again significantly outperforms the baselines for most of the metrics. Some of the baselines can also achieve good fidelity on certain models (e.g., UAD on OCSVM, 0.9847), but their performance is extremely unstable for other models (e.g., UAD on AE, 0.0045). It shows that the baseline methods cannot well serve as model-agnostic interpretation methods for unsupervised anomaly detectors, while our method can achieve much more stable performance on different models. Therefore, it is concluded that our method can obtain rules of high quality from different black-box unsupervised anomaly detection models using only unlabelled one-class data.

---

> > > ### Author Response · Authors · 2023-08-18
> > > **Response to Your Questions (Part 3)**
> > >
> > > ### About the Difference between OC-Tree and Our Method
> > >
> > > We greatly appreciate your in-depth investigation on our method and the related work. As the reviewer mentioned:
> > >
> > > > The density estimate often can be used as a proxy for the anomaly score and could be considered as the output of the anomaly detector.
> > >
> > > Though this point is not the initial idea of the OC-Tree, we think it does make it possible for OC-Tree to extract rules from unsupervised black-box models. Given the OC-Tree is not open-source, we made our best efforts to reproduce the OC-Tree by following its paper. Then, inspired by the reviewer’s idea, we incrementally revise it by translating the anomaly scores produced by black-box models into density estimates so that they can be utilized by the OC-Tree. We conducted a preliminary experiment to compare the performance of rule extraction from the AE model using the revised OC-Tree and our method on the two datasets. The results are as follows:
> > >
> > > |Method|Fidelity|Robustness|TPR|TNR|
> > > |-|-|-|-|-|
> > > |OC-Tree|0.5853|0.4586|0.999|0.1229|
> > > |Ours|0.9835|1.00|0.9457|0.9915|
> > >
> > > |Method|Fidelity|Robustness|TPR|TNR|
> > > |-|-|-|-|-|
> > > |OC-Tree|0.0994|0.4414|0.9845|0.0526|
> > > |Ours|0.9996|1.00|1.00 |0.9845|
> > >
> > > It can be seen that, though OC-Tree can achieve a fairly good detection rate of anomalies (i.e., TPR), it fails to reduce the false alarm rate to a satisfactory level (i.e., TNR), resulting in low scores of fidelity and robustness.
> > >
> > > This result is due to the following reasons. First, OC-Tree is inherently designed to exclude some training data points from normality if they have a relatively low density. Quoted from its paper (page 9):
> > >
> > > >Clipping KDE: $f_j(x)$ is thresholded at the level $\gamma\cdot \max_{x\in X}{f}_j(x)$.
> > >
> > > It means that those training data points that have scores lower than the maximum score multiplying a coefficient will be directly considered anomalous even if they are actually judged as normal by the black-box model. Consequently, this will increase the FPR of the OC-Tree.
> > >
> > > Second, when generating rules, OC-Tree chooses to shrink the rule bounds to strictly cover the training data points at a node, quoted from its paper (page 10):
> > >
> > > >Shrinking: The detected sub-intervals are shrunk in closed intervals in a way to fit the domain strictly covered by the related target training instances
> > >
> > > In this way, their generated rules will be more overfit to the training data points, and the space of normality defined by their rules will be small. However, the normal data points in the testing dataset could be somewhat different and located close to but outside the bounds of the rules. This explains why OC-Tree can achieve a good TPR while a much lower TNR for rule extraction.
> > >
> > > In contrast, our method first utilizes IC-Tree to achieve preliminary partitioning of the feature space; then, on top of each subspace defined by a leaf node, when generating rules, our method further exploits CBE algorithm to explore the decision boundary of the black-box model to determine the final extracted rules. This is highly important to an interpretation method for high fidelity, as the black-box model is likely to have **better generalization** so that their decision boundaries will not be accurately estimated by compact bounds. In fact, the OC-Tree is somewhat similar to the setting of *IC-Tree+Hypercube*, which had been proved to be inferior to our method in the ablation study.
> > >
> > > ### About Algorithmic Description
> > >
> > > We sincerely thank the reviewer for reading our provided algorithm and the suggestion for better clarification. We will adopt your advice in the final version of the paper!

---

> > > ### Author Response · Authors · 2023-08-18
> > > **Response to Your Questions (Part 4)**
> > >
> > > ### About Table 4
> > >
> > > Thanks for your suggestions. Indeed, considering that many previous works on model interpretation only focus on the presentation of numeric metrics (e.g., most of our baselines and [5-7]), we omitted many details due to the lack of space. Here is an overview of the steps that come up with the explanations:
> > >
> > > 1. For a reported anomaly $x$, we use the IC-Tree to pinpoint the rule of normality that judges $x$ as anomalous. It is realized by inputting $x$ into the tree and recursively finding the leaf node. The rule is denoted by $C_x$.
> > >
> > > 2. For each constraint of features in $C_x$, we compare the corresponding feature value of $x$ with the range of the constraint.
> > >
> > > 3. For the feature values outside the range of the constraints, the Rules of Normality, Feature Values, and Feature Meaning (i.e., the three columns in Table 4) along with the raw data sample will be sent as the explanation to the security expert for further analysis.
> > >
> > > 4. The security expert will analyze the data sample to determine the type of attack (i.e., the Attack column in Table 4) and give her/his understanding of the important attributes that make her/him identify the attack (i.e., the Human Understanding column in Table 4).
> > >
> > > 5. If there is a huge gap between the provided explanation and expert understanding, it indicates that the anomaly detector is not trustworthy. The security expert may conduct further actions to improve the detector, such as retraining, fine-tuning, or reconsidering feature selection.
> > >
> > > Note that such a gap may not only occur when the anomaly detector makes erroneous decisions but also when the anomaly detector correctly detects an attack sample by unreasonable features. For example, the most obvious attribute of a Denial-of-Service (DoS) attack is typically its high rate of forwarding packets in order to overwhelm the victim. However, the provided explanation might suggest the IP address is an important feature for the anomaly detector to make this inference, which is quite doubtful from the perspective of a security expert. In practice, this may be due to “spurious correlation” or “shortcut learning” (mentioned in [8] and [9]). Specifically, for the aforementioned DoS example, it could be due to inappropriate testbed settings for the collection of training data, such as the DoS attack is launched from a separate host address while all other normal traffic is from other host addresses.
> > >
> > > To improve the clarity of this subsection, we will add the process of explanation to the final version of the paper, as well as more details and examples in the appendix.
> > >
> > > Again, we really appreciate the replies from the reviewer, which are very helpful to improve our paper. If you still have any concerns, we are always happy to reply.
> > >
> > > [1] Hassan, Wajih Ul, et al. "Nodoze: Combatting threat alert fatigue with automated provenance triage." network and distributed systems security symposium. 2019.
> > >
> > > [2] O. Bastani, C. Kim, and H. Bastani, “Interpreting Blackbox Models via Model Extraction.” arXiv, Jan. 24, 2019.
> > >
> > > [3] Sarhan, Mohanad, Layeghy, Siamak, et al., “NetFlow Datasets for Machine Learning-based Network Intrusion Detection Systems”, arXiv, 2020
> > >
> > > [4] Liu, Lisa, et al. "Error Prevalence in NIDS datasets: A Case Study on CIC-IDS-2017 and CSE-CIC-IDS-2018." 2022 IEEE Conference on Communications and Network Security (CNS). IEEE, 2022.
> > >
> > > [5] Wu, Mike, et al. "Beyond sparsity: Tree regularization of deep models for interpretability." Proceedings of the AAAI conference on artificial intelligence. 2018.
> > >
> > > [6] Kazhdan, Dmitry, et al. "MEME: generating RNN model explanations via model extraction." arXiv preprint arXiv:2012.06954 (2020).
> > >
> > > [7] Li, Qiaomei, Rachel Cummings, and Yonatan Mintz. "Optimal Local Explainer Aggregation for Interpretable Prediction." Proceedings of the AAAI Conference on Artificial Intelligence. 2022.
> > >
> > > [8] A. S. Jacobs, et. al., “AI/ML for Network Security: The Emperor has no Clothes,” in Proceedings of the 2022 ACM SIGSAC Conference on Computer and Communications Security.
> > >
> > > [9] Arp, Daniel, et al. "Dos and don'ts of machine learning in computer security." 31st USENIX Security Symposium (USENIX Security 22). 2022.

---

> > > > ### Comment · Reviewer_mhUf · 2023-08-20
> > > > **Thanks for addressing my concerns**
> > > >
> > > > I thank the authors for addressing my concerns. I acknowledge the experimental rigor in responding to the reviews while also noting that the explanation part is not as strong as I would like. I will revise my score accordingly.

---

### Author Rebuttal · Authors · 2023-08-10

We thank the reviewers for their careful reading and detailed and considerate feedback.

We are glad that reviewers agree that our paper tackles an important problem (“an important problem domain” by mhUf, “tackles an open research problem” by AbmJ, “problem tackled in this paper is important” by jw73) and the proposed method demonstrates good experimental results (“outperform existing ones” by eS8h, “show improvement over baselines” by AbmJ, “works well in the experiments” by jw73).

The reviewers also mentioned that computational cost studies are necessary, and evaluation on other datasets would make the paper stronger. We are happy to report that we have conducted the majority of the suggested experiments (see attachments).

**Additional dataset.**
We have conducted additional experiments on an additional dataset (recommended by Reviewer AbmJ), which is an improved version of CIC-IDS dataset that we use in the paper. It consists of 80 features. Some of the flaws in the original CIC-IDS dataset are fixed (e.g., more accurate labeling), following the suggestions of Reviewer AbmJ. The average AUC scores of the four anomaly detection models on this dataset are as follow:

|AE|VAE|OCSVM|iForest|
|-|-|-|-|
|0.9906|0.9767|0.9901|0.9734|

As Table 1 in the attachment shows, our method still achieves the highest fidelity and TPR, for all anomaly detection models. Besides, our method also preserves a high level of robustness and TNR. The results indicate that our method can obtain rules of high quality from different black-box unsupervised anomaly detection models using only unlabeled one-class data.

**Computational cost.**
As suggested by reviewers, the discussions of computational costs and runtime performance are necessary to demonstrate the usability of the method, especially when the dimension of features is higher than the current feature numbers. Therefore, we add an experiment to measure the training time and prediction time. Since the improved CIC-IDS dataset has 80 features in total, we train the model using the first N features of 4000 samples, where N is set to 20, 40, 60, and 80. The results are shown in Table 2, which demonstrate the average training and prediction time of our method. It can be seen that the training time is around 1 minute, which is acceptable and practical for large-scale training. Besides, the training time increases basically linearly with the increase of feature numbers. This is because our method adopts a feature-by-feature strategy to explore the decision boundary of the model. For prediction time, our method is highly efficient, which only costs microsecond-level overhead for one inference. It shows that as a rule-based approach, our method can achieve real-time inference for online use.

We also compare the runtime performance to the baselines using the same data of 40 features, as shown in Table 3. We find that the training time of our method is basically at the same level as other methods. For prediction time, since all methods are rule-based, their execution is all at microsecond level, except for LIME which requires additional sampling for each inference. Note that the execution of our method is purely based on Python.  It is not yet optimized by C++ or Cython (as Trustee does). In practice, the prediction time of our method can be even higher with more efficient code implementation.

We also theoretically analyze the time complexity of our algorithms. For training process, the time complexity of the IC-Tree is identical to a CART: $O(d\cdot n\log n)$, where $d$ is the feature number and $n$ is the sample number; the time complexity of the CBE algorithm is $O(K\cdot d\cdot N_e\cdot N_s)$, where $K$ is the number of leaf nodes of the IC-Tree, $N_e$ and $N_s$ are the number of initial explorers and auxiliary explorers, respectively. Therefore, the training time is theoretically linear to the feature number, which is in line with the empirical results. For execution, the time complexity is $O(|C|\cdot d)$, where $|C|$ is the number of extracted rules.

Again, thank you for your comments, and we hope that this addresses your concerns! If you still have any questions or concerns, we are very willing to answer your questions or add other experiments during the discussion phase!

---

### Decision · Program_Chairs · 2023-09-21

**Decision:**

Accept (poster)

**Comment:**

This paper tackled an important question, explaining the decisions of complex anomaly detectors in a human-understandable way. The presented approach excited a deep discussion with reviewers and was able to convince even an initially very skeptical reviewer. As all reviewers eventually recommended the acceptance of this paper, the AC recommends acceptance.